

# Future shifts in extreme flow regimes in Alpine regions

Manuela I. Brunner[1], Daniel Farinotti[1,2], Harry Zekollari[1,2,3], Matthias Huss[2,4], and Massimiliano Zappa[1]

[1]Swiss Federal Institute for Forest, Snow and Landscape Research WSL, Birmensdorf ZH, Switzerland
[2]Laboratory of Hydraulics, Hydrology and Glaciology (VAW), ETH Zürich, Zürich, Switzerland
[3]Laboratoire de Glaciologie, Université Libre de Bruxelles, Brussels, Belgium
[4]Department of Geosciences, University of Fribourg, Fribourg, Switzerland

**Correspondence:** Manuela Brunner (manuela.brunner@wsl.ch)

**Abstract.** Extreme low and high flows can have negative economical, societal, and ecological effects and are expected to become more severe in many regions due to climate change. Besides low and high flows, the whole flow regime is subject to changes. Knowledge on future changes in flow regimes is important since regimes contain information on both extremes and conditions prior to the dry and wet season. Changes in individual low- and high-flow characteristics as well as flow regimes

under normal conditions have been thoroughly studied. In contrast, little is known about changes in extreme flow regimes. We here propose two methods for the estimation of extreme flow regimes and apply them to simulated discharge time series for future climate conditions in Switzerland. The first method relies on frequency analysis performed on annual flow duration curves. The second approach performs frequency analysis on the discharge sums of a large set of stochastically generated annual hydrographs. Both approaches were found to produce similar 100-year regime estimates when applied to a data set of

19 hydrological regions in Switzerland. Our results show that changes in both extreme low- and high-flow regimes for rainfall-dominated regions are distinct from those in melt-dominated regions. In rainfall-dominated regions, the minimum discharge of low-flow regimes decreases by up to 50%, whilst the reduction is of 25% for high-flow regimes. In contrast, the maximum discharge of low- and high-flow regimes increases by up to 50%. In melt-dominated regions, the changes point into the other direction than those in rainfall-dominated regions. The minimum and maximum discharge of extreme regimes increase by up to

100% and decrease by less than 50%, respectively. Our findings provide guidance in water resources planning and management and the extreme regime estimates are a valuable basis for climate impact studies.

**Keywords**: Stochastic simulation, PREVAH model, return period, frequency analysis, extreme value statistics, climate change

**Keypoints**:

1. Estimation of 100-year low- and high-flow regimes using annual flow duration curves and stochastically simulated discharge time series

2. Both normal and extreme regimes are changing under future climate conditions

3. The minimum discharge of extreme regimes will decrease in rainfall-dominated regions but increase in melt-dominated regions



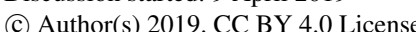


4. The maximum discharge of extreme regimes will increase and decrease in rainfall-dominated and melt-dominated regions, respectively

## 1 Introduction

Low flows can have severe impacts on ecology and economy. Potential ecological impacts include fish-habitat conditions or water quality (Rolls et al., 2012) whilst economical impacts comprise water supply, river transport, agriculture, and energy production (Van Loon, 2015). The intensity of such potentially harmful low flows is projected to increase in future due to climate change (Alderlieste et al., 2014; Papadimitriou et al., 2016; Marx et al., 2018). Also high flows, which can cause severe damages and major costs (Aon Benfield, 2016), are expected to change in future. While clear patterns of change have been detected for flood timing (Blöschl et al., 2017), changes in magnitude are less clear than for low flows (Madsen et al., 2014). Together with low and high flows, the whole flow regime, which depicts the magnitude, variability, and seasonality of discharge during the year (Poff et al., 1997), is expected to change (Arnell, 1999; Horton et al., 2006; Laghari et al., 2012; Addor et al., 2014; Milano et al., 2015). Such changes are caused by reduced snow and glacier storage (Beniston et al., 2018), related reductions in melt contributions (Farinotti et al., 2016; Jenicek et al., 2018), and changes in precipitation seasonality and intensity (Brönnimann et al., 2018). It is important to quantify these hydrological changes to adapt water governance and management accordingly (Clarvis et al., 2014).

Previous studies have focused on the detection of changes in mean flow regimes (Horton et al., 2006; Addor et al., 2014; Milano et al., 2015). For planning purposes and river basin management, however, not only estimates for normal conditions, but also, for extreme conditions are needed (Van Loon, 2015; Ternynck et al., 2016). Extreme regime estimates, which describe the evolution of flow over the year under extreme conditions, provide guidance for water managers, decision makers, and engineers involved in planning and water management. They are essential for the adaptation of hydraulic infrastructure such as reservoirs, and for developing suitable water management and flood protection strategies.

Commonly, extreme flow estimates focus on one characteristic of the hydrological regime, e.g. summer low flows, drought durations, drought deficits (e.g. Tallaksen, 2000; WMO, 2008), flood peaks, or flood volumes (e.g. Mediero et al., 2010; Brunner et al., 2017). The focus on one or several of these individual characteristics, however, neglects the pre-conditions of low- and high-flow events. Yet, for low-flow events, these pre-conditions are crucial for the formation of groundwater storage (Şen, 2015), reservoir filling (Hänggi and Weingartner, 2012; Anghileri et al., 2016), and soil moisture formation (Zampieri et al., 2009). These storages can become very important when it comes to the satisfaction of diverse water needs and to the alleviation of water shortages (Mussá et al., 2015; Brunner et al., 2019b). In the case of high flows, antecedent conditions determine the proportion of rainfall transformed to direct runoff and therefore the severity of the flood event (Berghuijs et al., 2016; Nied et al., 2017). In contrast to the individual low- and high-flow characteristics, the flow regime includes information on both the pre-conditions and the discharge during the low- and high-flow seasons.

Estimating extreme flow regimes with a given exceedance frequency is not straightforward since discharge values at several points in time are correlated. Because of the multivariate nature of the problem, no single solution exists. We here aim at




estimating extreme high- and low-flow regimes with a defined return period for current and future climate conditions. We propose two possible approaches for the estimation of such extreme regimes. The first approach is based on flow duration curves (FDCs). FDCs describe the whole distribution of discharge and are particularly suited for planning purposes (Vogel and Fennessey, 1994; Claps and Fiorentino, 1997). It has been shown that frequency analysis performed on annual FDCs allows for

the estimation of extreme FDCs with pre-defined return periods (Castellarin et al., 2004; Iacobellis, 2008). While such estimates contain information about the frequency of occurrence and the distribution of flow, they lack information on the seasonality of flow (Vogel and Fennessey, 1994). FDC estimates derived for a certain return period $T$ therefore need to be recombined with a specific seasonality, e.g. the long-term one. This first estimation approach treats distribution and seasonality separately. To overcome this problem, an alternative approach based on stochastically generated time series is proposed. Stochastically

generated time series have been used in a number of water resources studies, including hydrologic design and drought planning (Koutsoyiannis, 2000). Stochastic approaches generate large sets of possible discharge time series, thus sampling hydrologic variability beyond the historical record (Herman et al., 2016; Tsoukalas et al., 2018), potentially including extreme events and regimes. In hydrology, stochastic models have been developed so as to reproduce key statistical features of observed data, including the distribution and the temporal dependence (Sharma et al., 1997; Salas and Lee, 2010; Tsoukalas et al., 2018).

Traditionally, indirect approaches which combine the stochastic simulation of rainfall with hydrological models, have been used for the generation of stochastic discharge time series (Pender et al., 2015). Direct approaches stochastically simulate discharge. The simplest type of models to describe daily streamflow are autoregressive moving average (ARMA) models (Pender et al., 2015; Tsoukalas et al., 2018). However, this type of models only captures short-range dependence (Koutsoyiannis, 2000). Models also capturing long-range dependence include fractional Gaussian noise models (Mandelbrot, 1965), fast

fractional Gaussian noise models (Mandelbrot, 1971), broken line models (Mejia et al., 1972), and fractional autoregressive integrated moving average models (Hosking, 1984). An alternative to these time-domain models are frequency domain models (Shumway and Stoffer, 2017). These latter use phase randomization to simulate surrogate data with the same Fourier spectra as the raw data (Theiler et al., 1992; Radziejewski et al., 2000). Despite their favorable characteristics, such methods based on the Fourier transform have been rarely applied in hydrology (Fleming et al., 2002). We apply the approach of phase randomization

in combination with the flexible Kappa distribution (Hosking, 1994) to simulate stochastic discharge time series as proposed by Brunner et al. (2019a). Among these simulated series, extreme regimes can be identified. After having identified a suitable approach for the estimation of extreme regimes, we apply this approach to discharge time series representing future climate conditions. A comparison to current estimates allows us to identify future changes in extreme high- and low-flow regimes.

## 2  Methods

### 2.1  Study area

The analyses were performed on a set of 19 hydrological regions in Switzerland (Figure 1) with areas between 600 and 5000 km$^2$, mean elevations between 550 and 2300 m a.s.l., and mean annual precipitation sums between 1000 and 1800 mm. The flow regimes north of the Alps (Plateau and Jura) are dominated by rainfall and characterized by high discharge in winter and



spring but low discharge in summer. In contrast, the regimes in the Alps are dominated by snow and ice melt and characterized by high discharge in summer. For illustration purposes, we chose four regions. Two of them (Jura and Thur) have a rainfall-dominated regime and the other two (Valais and Engadin) a melt-dominated regime.

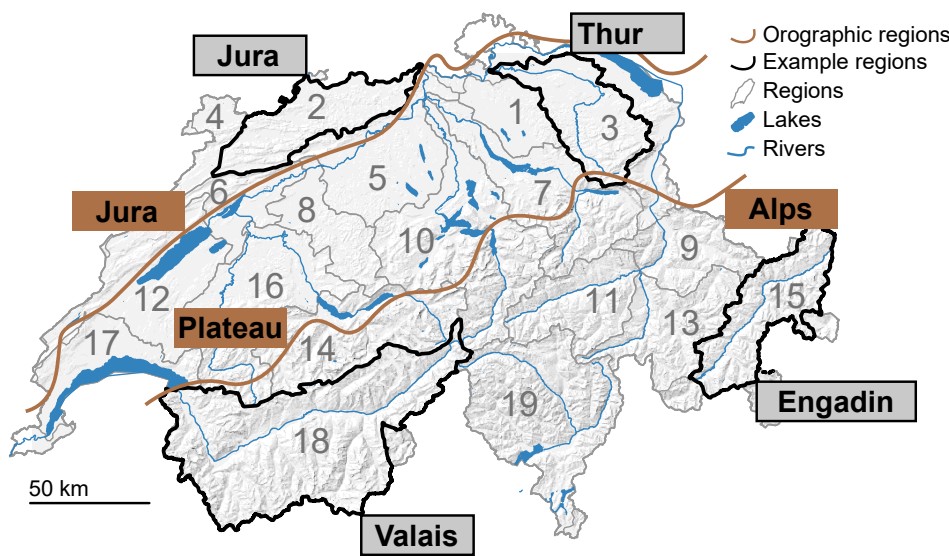

**Figure 1.** Map of Switzerland with 19 large hydrological regions (grey outline) and the four illustration regions (black border): Thur, Jura, Valais, and Engadin. The main orograpic regions Jura, Plateau, and Alps are outlined by the brown lines.

## 2.2 Analysis framework

5    The analysis performed to detect changes in future extreme regimes consisted of three main steps (Figure 2). First, different procedures for estimating extreme flow regimes were tested. Once a suitable procedure was identified, it was applied to estimate extreme high- and low-flow regimes under current and future climate conditions (second step). These extreme regimes were compared to mean regimes. Third, current and future estimates were compared to detect future changes in flow regimes. We used simulated discharge representing both current and future climatology as the basis for the analysis. The current discharge

10    series were derived by feeding a hydrological model with observed meteorological data. The future discharge series were obtained by driving the model with meteorological data from climate model simulations.

    The two estimation techniques applied use frequency analysis on different quantities. The first method applies frequency analysis to the individual percentiles of the FDC. The second method uses stochastically simulated discharge time series to identify annual hydrographs with a certain non-exceedance probability. We refer to these methods as *FDC* and *stochastic*,

15    respectively. The two methods are compared to a benchmark method (*univariate*), which performs univariate frequency analysis on the monthly discharge values and neglects the dependence between individual months. We here focus on the estimation of high- and low-flow regimes with a return period of $T = 100$ years since this return period is commonly used for planning purposes. The methods outlined in this study, however, can be generalized to other return periods. In the following paragraphs,





we describe the data sets (Figure 2 A, Sect. 2.3), the stochastic discharge generation procedure (Figure 2 B, Sect. 2.4), and the estimation techniques used to derive extreme flow regimes (Figure 2 C, Sect. 2.5).

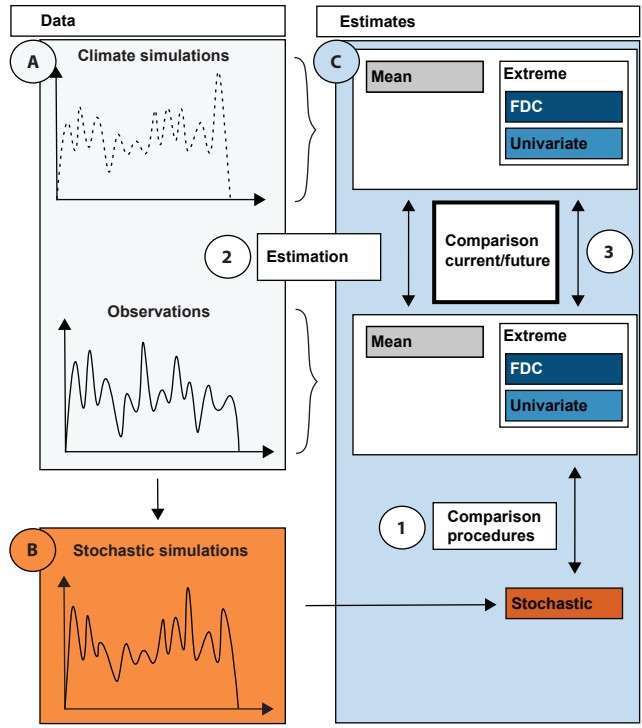

**Figure 2.** Illustration of study framework: 1) Comparison of different estimation techniques *univariate*, *FDC*, and *stochastic*, 2) estimation of current and future mean and extreme regimes using simulated discharge time series, and 3) comparison of current and future regime estimates. The manuscript A) introduces the simulated data used, B) outlines the stochastic discharge generator, and C) describes the estimation approaches.

## 2.3 Hydrological simulations

We used discharge time series simulated with the hydrological model PREVAH (Viviroli et al., 2009b) as input for the analysis. To represent current conditions, the model was driven with observed meteorological data for the period 1981-2010. To represent future conditions, it was driven with meteorological data obtained by regional climate model simulations for the period 2071-2100 (see below). PREVAH is a conceptual process-based model. It consists of several sub-models representing different parts of the hydrological cycle: interception storage, soil water storage and depletion by evapotranspiration, groundwater, snow accumulation and snow- and glaciermelt, runoff and baseflow generation, plus discharge concentration and flow routing (Viviroli et al., 2009b). A gridded version of the model at a spatial resolution of 200 m was set up for Switzerland (Speich et al., 2015). For the calibration of the model parameters, meteorological and discharge time series from 140 mesoscale catchments covering different runoff regimes were used. The model calibration was conducted over the period 1993–1997. Validation on discharge was performed with the period 1983–2005. More details on the calibration and validation procedures can be found





in Köplin et al. (2010). The parameters for each model grid cell were derived by regionalizing the parameters obtained for the 140 catchments with ordinary kriging (Viviroli et al., 2009a; Köplin et al., 2010). Future glacier extents were simulated with two glacier evolution models. We used the global glacier evolution model (GloGEM; Huss and Hock, 2015) for short glaciers (glacier length < 1 km) and GloGEMflow (Zekollari et al., 2018) for long glaciers (length > 1 km). GloGEM simulates glacier

changes with a retreat parameterization relying on observed glacier changes (Huss et al., 2010). GloGEMflow is an extended version of GloGEM with a dynamic ice flow component. This new model was extensively validated over the European Alps, through comparisons with various observations (e.g. surface velocities and observed glacier changes) and detailed 3D projections from modelling studies focusing on individual glaciers (e.g. Jouvet et al., 2011; Zekollari et al., 2014). The simulated glacier extents were transformed from the GloGEM(flow) 1D model grid to the 2D PREVAH model grid by ensuring that the

area for each elevation band was conserved.

PREVAH is driven by time series of precipitation, temperature, relative humidity, radiation, and wind speed. The meteorological forcing for current simulations was observed time series provided by the Federal Office of Meteorology and Climatology MeteoSwiss (2018) while the transient meteorological forcing for future simulations was derived from the CH2018 climate scenarios (National Centre for Climate Services, 2018). They are based on the results from the EURO-CORDEX initiative (Ja-

cob et al., 2014; Kotlarski et al., 2014), which are the most sophisticated and high-resolution coordinated climate simulations over Europe. The scenarios are based on representative concentration pathways (RCP) (Moss et al., 2010; Meinshausen et al., 2011; van Vuuren et al., 2011) and a regional downscaling approach based on quantile mapping (Themeßl et al., 2012; Gudmundsson et al., 2012). The meteorological data were derived from an ensemble of 39 model chains (Table A1 in Appendix), which provide temperature, precipitation, relative humidity, radiation, and wind speed, for various meteorological stations in

Switzerland. The meteorological data were interpolated to a $2 \times 2$ km grid using detrended inverse distance weighting where the detrending was based on a regression between climate variables and elevation (Viviroli et al., 2009b). During a model run, PREVAH reads the meteorological grids and further downscales the data to the computational grid of $200 \times 200$ m using bilinear interpolation. For temperature, a lapse rate of -0.65 °C/100m was additionally used.

## 2.4 Stochastic simulation of discharge time series

The discharge simulated with the hydrological model for the current (1981–2010) and future (2071–2100) 30-years periods only represent small sets of possible annual hydrograph realizations. Among these realizations, certain hydrographs including extreme hydrographs such as a 100-year hydrograph were possibly not observed. We used a stochastic discharge simulation procedure to increase the number of possible annual hydrograph realizations. These realizations represent the discharge statistics and temporal correlation structure of the available data, and extend the existing sample to yet unobserved annual hydrographs.

To simulate such hydrographs, we used the method of *phase randomization* (Theiler et al., 1992; Schreiber and Schmitz, 2000). We combined this empirical procedure with the flexible four-parameter Kappa distribution (Hosking, 1994) to allow for the extrapolation to yet unobserved values. This phase randomization approach preserves the autocorrelation structure of the raw series by conserving its power spectrum (Theiler et al., 1992). The procedure consists of three main steps (Radziejewski et al., 2000). In a first step, the discharge series is converted from the time domain to the spectral domain by Fourier transformation





(Morrison, 1994). The Fourier transformation of a given time series $x = (x_1, \ldots, x_t, \ldots, x_n)$ of length $n$ is

$$f(\omega) = \frac{1}{\sqrt{2\pi n}} \sum_{t=1}^{n} e^{-i\omega t} x_t, \quad -\pi \leq \omega \leq \pi, \tag{1}$$

where $t$ is the time step and $\omega$ are the phases. In this spectral domain, the data are represented by the phase angle and by the amplitudes of the power spectrum as represented by the periodogram. The phase angle of the power spectrum is uniformly
distributed over the range $-\pi$ to $\pi$. In a second step, the phases in the phase spectrum are randomized while the power spectrum is preserved. In a third step, the inverse Fourier transform is applied to transform the data from the spectral domain back to the temporal domain. A step by step description of the stochastic simulation procedure and more background information on the Fourier transform are provided in Brunner et al. (2019a) and references therein.

An application of the simulation procedure to four example catchments in Switzerland has shown that both seasonal statistics
and temporal correlation structures of discharge can be well reproduced (Brunner et al., 2019a). We therefore used this method to stochastically simulate 1500 years of discharge for each of the 19 regions in our data set. Stochastic series representing current conditions were generated by using the hydrological model simulations for 1981–2010 as input. Stochastic series representing future conditions were generated based on each of the hydrological model simulations generated with the 39 model chains.

## 2.5   Estimation of $T$-year hydrographs

We employed two methods for estimating 100-year low- and high-flow regimes: *FDC* and *stochastic*. The extreme regime estimates were compared to the stochastically generated hydrographs to check for plausibility. Furthermore, they were compared to a lower-bound (for low-flow regimes) or upper-bound (for high-flow regimes) benchmark regime derived by combining 100-year monthly discharge estimates obtained from univariate frequency analysis. This frequency analysis was performed on the
values of each month independently and the monthly values were fitted with a Generalized Extreme Value (GEV) distribution. This distribution was not rejected according to the Anderson–Darling goodness-of-fit test computed using the procedure proposed by Chen and Balakrishnan (1995) ($\alpha = 0.05$). The disadvantage of the univariate procedure is that the autocorrelation in the data, which is mainly visible for lags of 1 and 2 months, is neglected, which overestimates the extremeness of the 100-year low-flow regime and therefore produces unrealistic estimates.

### 2.5.1   FDC

A first extreme regime estimate was derived by performing the frequency analysis on annual FDCs. According to Vogel and Fennessey (1994), an annual FDC with an assigned return period can be obtained from the $p^{\text{th}}$ quantile function. To do so, we fitted a GEV distribution to the quantiles corresponding to each percentile. The GEV was not rejected based on the Anderson–Darling goodness-of-fit test ($\alpha = 0.05$). The fitted GEV distributions were used to estimate the 100-year quantile
for each percentile. The 100-year FDC was then derived by combining these 100-year quantiles. The 100-year FDC does not contain any information about the seasonality but only about the statistical distribution of flow. To include information about



seasonality, we combined the estimated 100-year FDC with a typical seasonal regime. To do so, the individual quantile values of the FDC were assigned to the corresponding ranks of a typical flow regime. This typical regime was defined as the long-term (mean) regime of the input time series.

### 2.5.2 Stochastic

5   The second method for the estimation of extreme regimes performs the frequency analysis directly on a large set of stochastically simulated annual hydrographs (1500 years). The frequency analysis was performed on the annual sums of the stochastically generated hydrographs. We identified the hydrograph corresponding to the empirical 100-year annual discharge sum as the 100-year regime. The application of this procedure is only possible for long time series as given by the stochastic series, since a 100-year annual sum is not necessarily observed in a short record of, say, 30 years.

### 10  2.6   Comparison of current and future regime estimates

The two methods and the benchmark approach for the estimation of 100-year low- and high-flow regime estimates were applied to discharge time series representing current and future climate conditions. First, 100-year regimes were estimated for current conditions (1981–2010). For generating a *control* regime, we used the discharge simulated with the observed meteorological data. For representing uncertainty due to different climate model chains, we derived one *reference* regime for each discharge

15  time series simulated by the 39 climate model chains. This analysis provided us with a range of current regime estimates due to climate model uncertainty. In a second step, 100-year estimates were derived for future conditions using the simulated time series for the period 2071–2100 for all model chains. Model chains belonging to different RCPs were treated separately. We assessed changes in seasonality and magnitude of flow regimes in terms of their minimum, maximum, and mean discharge (Figure 3).

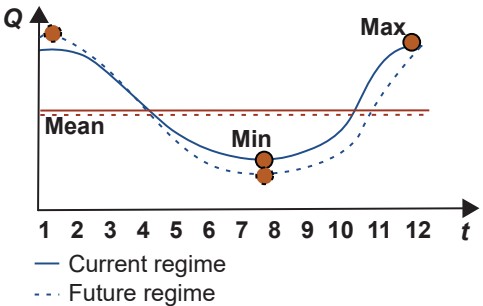

**Figure 3.** Main characteristics of an annual rainfall-dominated flow regime under current and future conditions: maximum, mean, and minimum.




## 3 Results

### 3.1 Comparison of estimation methods

The two estimation techniques and the benchmark approach provide distinct estimates for the 100-year low-flow regimes (Figure 4). The univariate technique leads to the most extreme regimes whilst the FDC and stochastic methods lead to similar estimates. The univariate estimate should only be seen as a lower benchmark and not as an estimate for a "true" 100-year regime since the univariate approach neglects the dependence between monthly estimates. In contrast, the FDC and stochastic approaches produce more plausible estimates, i.e., estimates at the lower bound of the observed values. The summer low-flow regimes estimated by the FDC technique are comparable to the regimes of the year 2003, which included a very dry summer (Beniston, 2004; Rebetez et al., 2006; Schär et al., 2004; Zappa and Kan, 2007).

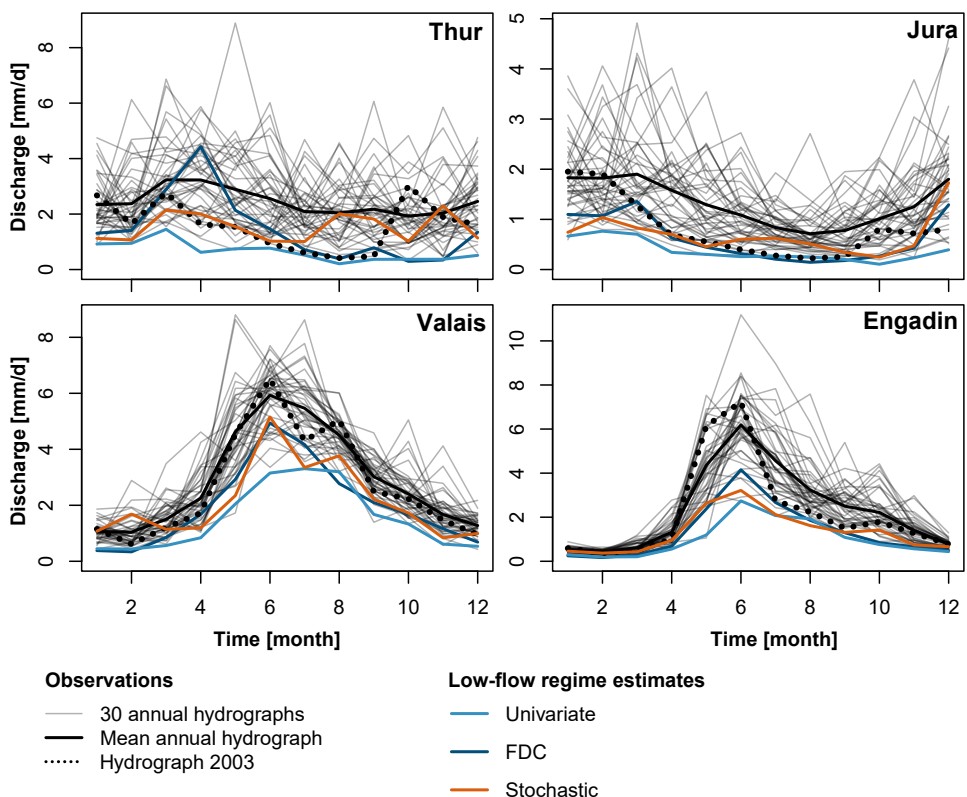

**Figure 4.** 100-year **low-flow** regime estimates for current climate conditions (control) derived using univariate frequency analysis (light blue), frequency analysis on the FDC (dark blue), and stochastically generated time series (orange). The observed annual hydrographs are given in grey while the mean annual hydrograph and the hydrograph of the year 2003 are given in black. The four panels are shown on different scales.





Similar as for low-flow regimes, the 100-year high-flow regimes derived by the three estimation techniques are distinct (Figure 5). The univariate estimate is not realistic and therefore serves as an upper benchmark. The FDC and stochastic techniques produce more realistic estimates, at the upper bound of the observed annual hydrographs. Contrary to low-flow estimates, high-flow estimates generated with the FDC or stochastic technique can be different. The stochastic approach generally leads

5  to more conservative estimates than the FDC approach in melt-dominated regions. We attribute this to the fact that the stochastic approach performs frequency analysis on annual sums while the FDC approach performs frequency analysis on the percentiles of the FDC.

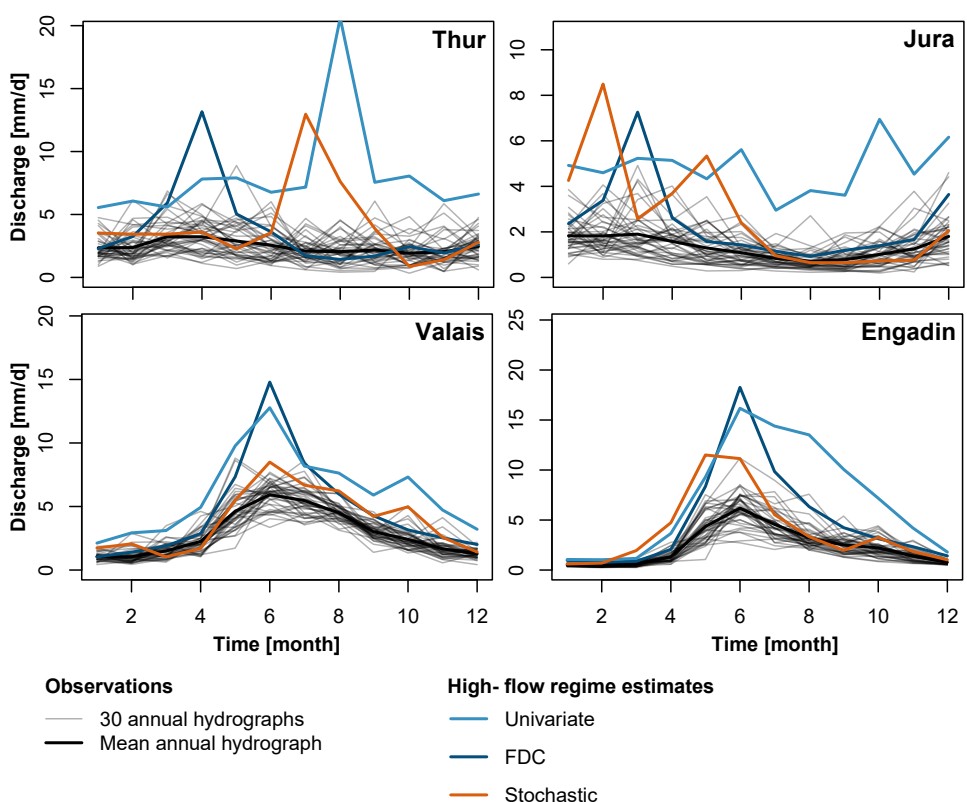

**Figure 5.** 100-year **high-flow** regime estimates for current conditions (control) derived using univariate frequency analysis (light blue), frequency analysis on the FDC (dark blue), and stochastically generated time series (orange). The observed annual hydrographs are given in grey while the mean annual hydrograph is given in black. The four panels are shown on different scales.

The plausibility of the 100-year estimates derived by using the FDC and stochastic approaches is shown by a comparison with stochastically generated annual hydrographs (Figure A1 in the Appendix for the low-flow estimates). The derived estimates,

10  in fact, are embedded in the lower spectrum of the stochastically generated annual hydrographs. This is hardly the case for the univariate estimates, which lead to "unrealistically low" 100-year hydrographs partly outside of the range of the stochastically generated hydrographs. Similarly, the 100-year high-flow regime estimates derived by the FDC and stochastic methods are



embedded in the higher spectrum of the stochastically generated hydrographs while the univariate estimate is "unrealistically high". Since the univariate approach yields unrealistic estimates, it is not considered for further analysis.

## 3.2 Current and future low-flow regime estimates

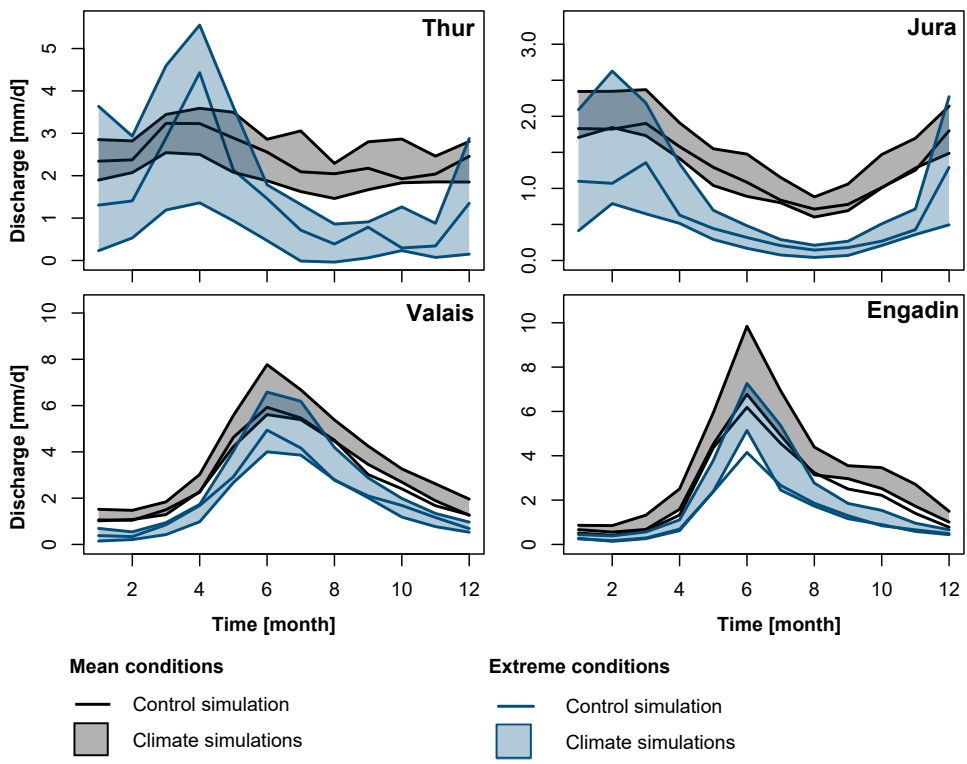

**Figure 6.** Current 100-year normal (grey) and **low-flow** regimes (blue) estimated by using the FDC method on the control discharge simulations derived by observed meteorological data (bold line) and the reference discharge simulations derived by meteorological data simulated by the 39 model chains for the reference period (shaded polygons).

Both normal and extreme regimes are subject to uncertainty when derived from simulated discharge. The uncertainty comes
5  from the hydrological model and from the spread between the climate model chains. Figure 6 shows normal and extreme low-flow regime estimates derived for the observed climatology for the four illustration catchments. It also shows the range of regimes obtained by using different model chains. This range of regimes generally encompasses the regime derived from observed data, which means that the climate model output realistically reproduces the observed climate. An exception is the Engadin, where the regimes derived from the model chains overestimate summer low flows. This is due to an overestimation
10  of precipitation. The spread in the current regimes is larger for extreme than for normal conditions for the rainfall-dominated catchments Thur and Jura. This range should be kept in mind when analyzing future regime estimates.




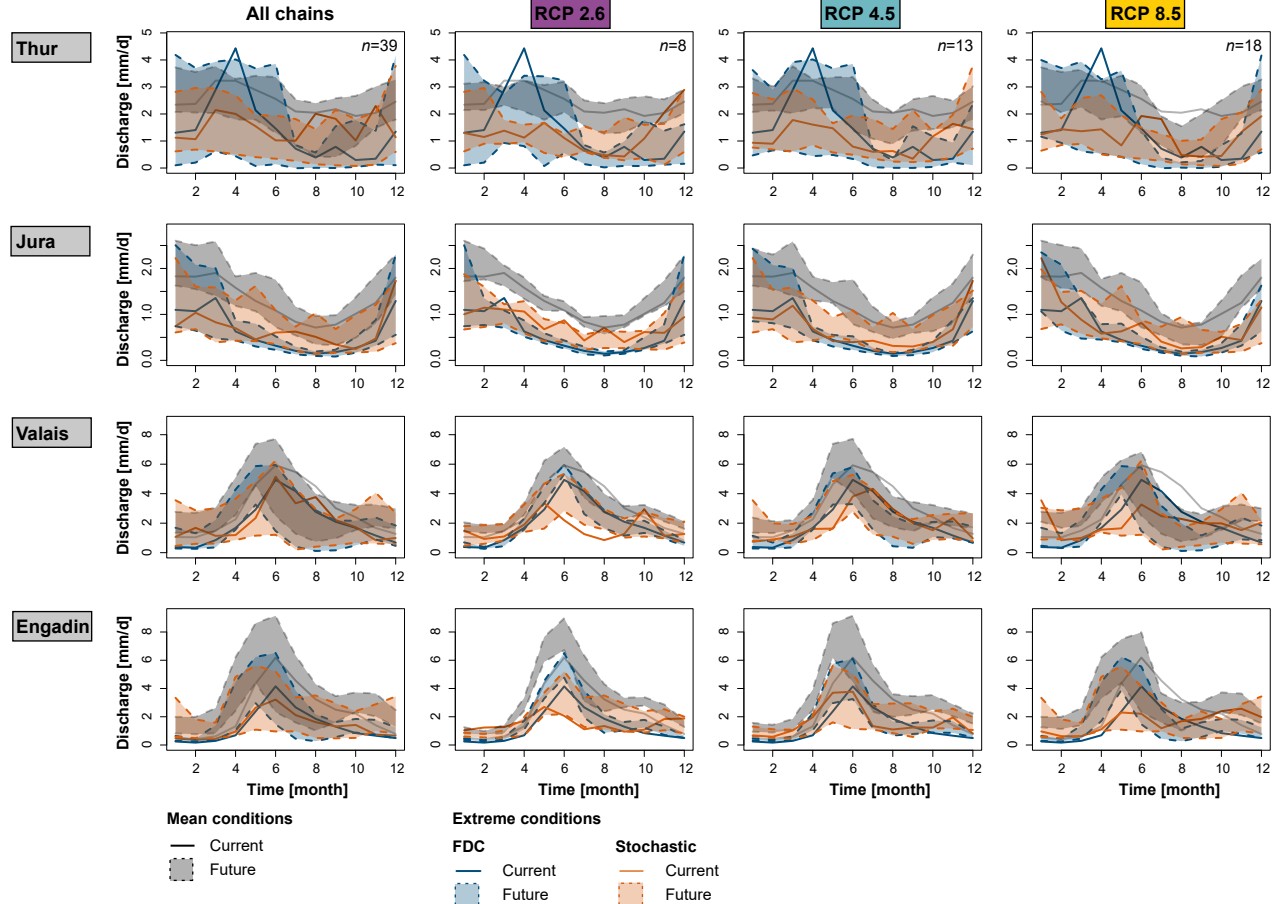

**Figure 7.** Comparison of current (solid line) and future 100-year **low-flow** regime estimates (shaded polygons) over the 39 model chains derived by the FDC (blue) and stochastic (orange) approaches. The normal regimes are provided as a reference (grey).

Shifts in regimes are expected for both normal and for extreme low-flow conditions (Figure 7). The shifts are weak for rainfall-dominated regions (e.g. Thur and Jura), while they are strong for melt-dominated regions (e.g. Valais and Engadin). For the rainfall-dominated regions, changes in normal and extreme regimes are only visible for RCP 8.5. These chains lead to regimes with more expressed summer low flows. In the case of melt-dominated regions, most model chains lead to clear shifts
5   towards regimes with earlier and reduced summer flows. These shifts are more pronounced for RCP 8.5 than RCPs 4.5 and 2.6. Note that the spread of future regimes is smaller for RCP 2.6 than RCPs 4.5 and 8.5 due to the smaller number of chains in the ensemble.

Differences in current and future mean and extreme low-flow regimes are summarized in Figure 8. The detected changes for RCP 2.6 and RCP 8.5 are similar. Changes are projected for the minimum and maximum discharge of mean and extreme
10   low-flow regimes and for their timing, but less for the mean of these regimes. The changes of the mean flow can reach up to 50%, while the maximum and mimimum flows can change up to 100%.



Changes in melt- and rainfall-dominated regions are clearly distinct. In rainfall-dominated regions, an increase is expected in the mean regime. The FDC approach also suggests changes in extreme regimes, whilst such changes are less pronounced in the low-flow estimates of the stochastic approach. In contrast, a decrease is expected in the discharge minimum, independent of the considered estimation technique. For melt-dominated regions, an opposite change pattern is found. There, a decrease

5  in maximum discharge and an increase in minimum discharge is expected for both normal and extreme low-flow regimes. Shifts of one or two months are expected in timing for both rainfall- and melt-dominated regions. In most catchments, the timing of future maximum discharge is likely to occur earlier than under current conditions. Shifts towards later in the year are expected in the timing of the minimum flow. The changes in mean and minimum flows are similar for extreme low-flow regimes derived by the two estimation techniques FDC and stochastic. In contrast, the maximum flow and the timing are more

10  variable if applying the stochastic instead of the FDC approach.



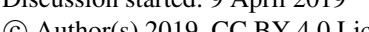

**Figure 8.** Differences between current and future normal (grey) and extreme **low-flow** regime characteristics for the 19 regions (Figure 1) estimated by the FDC (blue) and stochastic (orange) approaches. Five indicators are shown: minimum discharge, maximum discharge, mean discharge, timing of minimum discharge, timing of maximum discharge. The first three rows show relative changes, the second three rows show changes in months. Melt-dominated (dark colors) and rainfall-dominated regions (light colors) are distinguished. The boxplots indicate the range resulting from using the 39 model chains.

### 3.3 Current and future high-flow regime estimates

High-flow regime estimates are also expected to change (Figure 9). Changes in high-flow extreme regimes are slightly more pronounced for RCP 8.5 than for RCP 2.6 (Figure 10). They are independent of the estimation technique used (FDC/stochastic). As for the low-flow regimes, only moderate and mostly positive changes of less than 30% are expected in the mean discharge of extreme high-flow regimes. The changes in the maximum and minimum discharge of the high-flow regimes are much stronger,



i.e up to 100%. In rainfall-dominated regions, changes in maximum discharge are mostly positive while they can be negative for melt-dominated regions. In these melt-dominated regions, an increase is expected in the minimum discharge of high-flow extreme regimes. In rainfall-dominated regions, changes in minimum discharge can also be negative, especially for RCP 8.5.

Changes in timing are more pronounced when using the stochastic technique and less when using the FDC technique. However, there is no consistent pattern across catchments. Minimum and maximum discharge can occur earlier or later in the year than under current conditions.

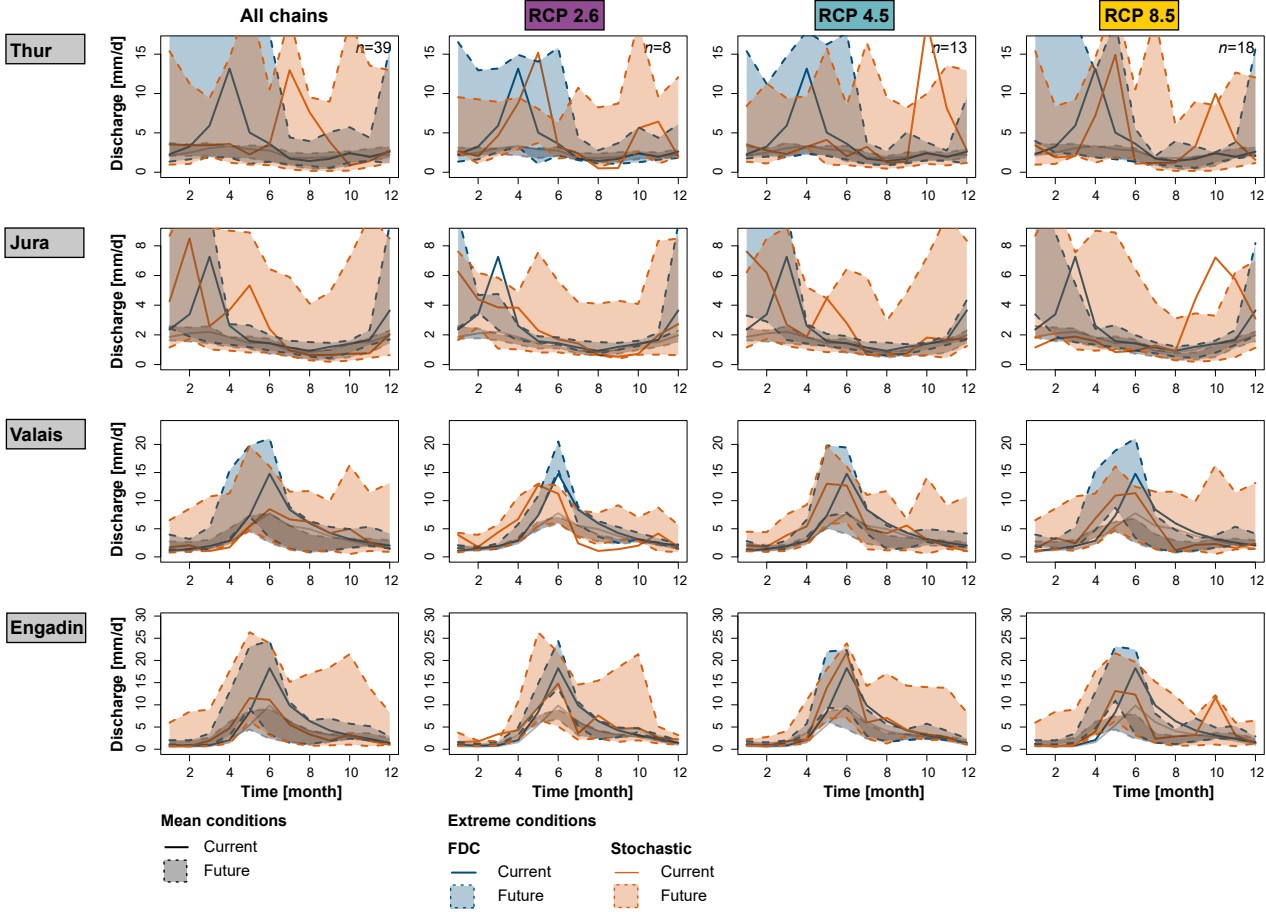

**Figure 9.** Comparison of current (solid line) and future 100-year **high-flow** regime estimates (shaded polygons) over the 39 model chains derived by the FDC (blue) and stochastic (orange) approaches. The normal regimes are provided as a reference (grey).



**Figure 10.** Differences between current and future normal (grey) and extreme **high-flow** regime characteristics for the 19 regions (Figure 1) estimated by the FDC (blue) and stochastic (orange) approaches. Five indicators are shown: minimum discharge, maximum discharge, mean discharge, timing of minimum discharge, timing of maximum discharge. The first three rows show relative changes, the second three rows show changes in months. Melt-dominated (dark colors) and rainfall-dominated regions (light colors) are distinguished. The boxplots indicate the range resulting from using the 39 model chains.



## 4 Discussion

### 4.1 Estimation methods

The low-flow regime estimates derived with the univariate method are implausible because the method neglects the interdependence between flows of adjacent months. In contrast, both other methods, FDC and stochastic, lead to similar results. These estimates seem to be plausible in the light of the stochastically generated hydrographs, which represent a large set of possible realizations among which extreme hydrographs can be found. While the estimates derived by the two methods do not differ much, both methods have their advantages and disadvantages. The FDC approach is relatively simple to implement, but decouples seasonality from the distribution of daily discharge values. In contrast, the stochastic approach jointly considers magnitude and seasonality, but requires the implementation of a stochastic discharge generator. The main advantage of such a generator is that the individual hydrograph realizations can be used for specific impact studies, which allows for directly performing the frequency analysis on the quantity of interest. There are several possible solutions to the multivariate problem of estimating extreme regimes, and none of these two methods can therefore be said to be the better one.

The estimation of extremes, be it of regimes or individual flow characteristics, is associated with several sources of uncertainty. These comprise the choice of an extreme value distribution used to fit the data (i.e. percentiles of FDCs, annual sums, daily discharge sums) and the estimation of its parameters (Merz and Thieken, 2005; Brunner et al., 2018a). When applied to time series representing future conditions simulated with a hydrological model, additional uncertainty sources are involved. These include future climate scenarios, global climate model structures, initial conditions, downscaling methods, modelled future glacier extents, the choice of a hydrologic model and the calibration of its parameters (Wilby et al., 2008; Addor et al., 2014; Clark et al., 2016). Despite these uncertainties, the extreme regime estimates can be used to identify future changes and as such these estimates can be further used in climate impact studies. Potential fields of application include water scarcity assessments, where such regime estimates are combined with estimates on water demand (Brunner et al., 2018b), eco-hydrological studies (Wood et al., 2008), or analyses on the future potential of hydropower production (Schaefli et al., 2019).

### 4.2 Changes in future regime estimates

Changes in all types of regimes (normal/extreme low flow/extreme high flow), were found to be distinct for melt-dominated and rainfall-dominated regions. This refers not only to the entire regime, but also to individual regime characteristics such as minimum, maximum, and mean flow as well as the timing of the minimum flow. The direction of change was different in rainfall- and melt-dominated regions for all regime types. An increase of up to 50% in the maximum discharge of normal and extreme low- and extreme high-flow regimes was found for rainfall-dominated regions. In contrast, a decrease of the minimum discharge by up to 100% is projected to occur for these catchments and all types of regimes. The opposite is true for melt-dominated regimes, where the minimum discharge increases while the maximum and mean discharge decrease. The changes in extreme regimes can be explained by a reduction or an earlier contribution of snow- and glacier melt (Hanzer et al., 2018), and by an increase in winter precipitation (Jenicek et al., 2018), which coincides with the high-flow season in rainfall-dominated regions but the low-flow season in melt-dominated regions. For normal regimes, changes in melt-dominated regimes were




found in previous studies (Barnett et al., 2005; Jenicek et al., 2018; Fatichi et al., 2014; Hanzer et al., 2018). Fatichi et al. (2014) found a projected discharge decrease in melt-dominated regions due to reduced contribution of ice melt in the Po and Rhine river basins. The regime shifts in the rainfall-dominated regions are also influenced by increases in precipitation in the winter season and decreases in the summer season. Precipitation increases in the high-flow winter season lead to increases in the

discharge maximum, while precipitation decreases in the low-flow summer season lead to decreases in the discharge minimum. The results of Fatichi et al. (2014) confirm that changes in rainfall-dominated regions are more uncertain since the projected changes in precipitation mostly lie within the range of natural variability of the control scenario. Similar results were found by Jenicek et al. (2018) for several catchments in Switzerland and by Barnett et al. (2005) on a global scale. We here showed that these previous findings also apply to extreme regimes. The regime shifts detected have implications for various sectors.

Regime shifts and more severe low flows were found to lead to more severe water scarcity situations, where water supply is insufficient to meet water demand (Brunner et al., 2019b). In the hydropower sector, future regime shifts are anticipated to lead to a reduction in production (Finger et al., 2012; Schaefli et al., 2019).

## 5   Conclusions

Extreme regime estimates were derived by frequency analysis performed on 1) annual flow duration curves (FDCs) and 2) on

the discharge sums of stochastically generated annual hydrographs. Both were found to provide realistic, mutually-agreeing results. A range of future extreme regime estimates was obtained both for extreme and normal conditions. In rainfall-dominated regions, the range of these future low- and high-flow estimates comprised the current estimate. On the contrary, in melt dominated regions, future high-flow and especially low-flow regimes were distinct from the current estimate. Changes in mean discharges were moderate for all types of regimes and catchments, and did not exceed 50%. Projected changes in the mini-

mum discharge of normal and extreme high- and low-flow regimes were positive in melt-dominated regions due to increases in winter precipitation and amount to up to a 100%. In contrast, mostly positive changes, of up to 50% in maximum discharge were found in rainfall-dominated regions for all types of regimes. These positive changes in maximum discharge are linked to increases in winter precipitation, which coincides with the high flow season. High- and low-flow regime estimates derived using the approaches proposed in this study are important for climate impact studies addressing e.g. the future hydropower

production potential or the occurrence of water shortage situations. The estimates also provide guidance for hydraulic design, emergency planning and drought and water management.

*Data availability.* The climate model simulations are available on the webpage of the Swiss National Centre for Climate Services (https://www.nccs.admin.ch/nccs/de/home.html). The hydrological model simulations are available upon request from Massimiliano Zappa (massimiliano.zappa@wsl.ch). The extreme regime estimates are available upon request from Manuela Brunner (manuela.brunner@wsl.ch).



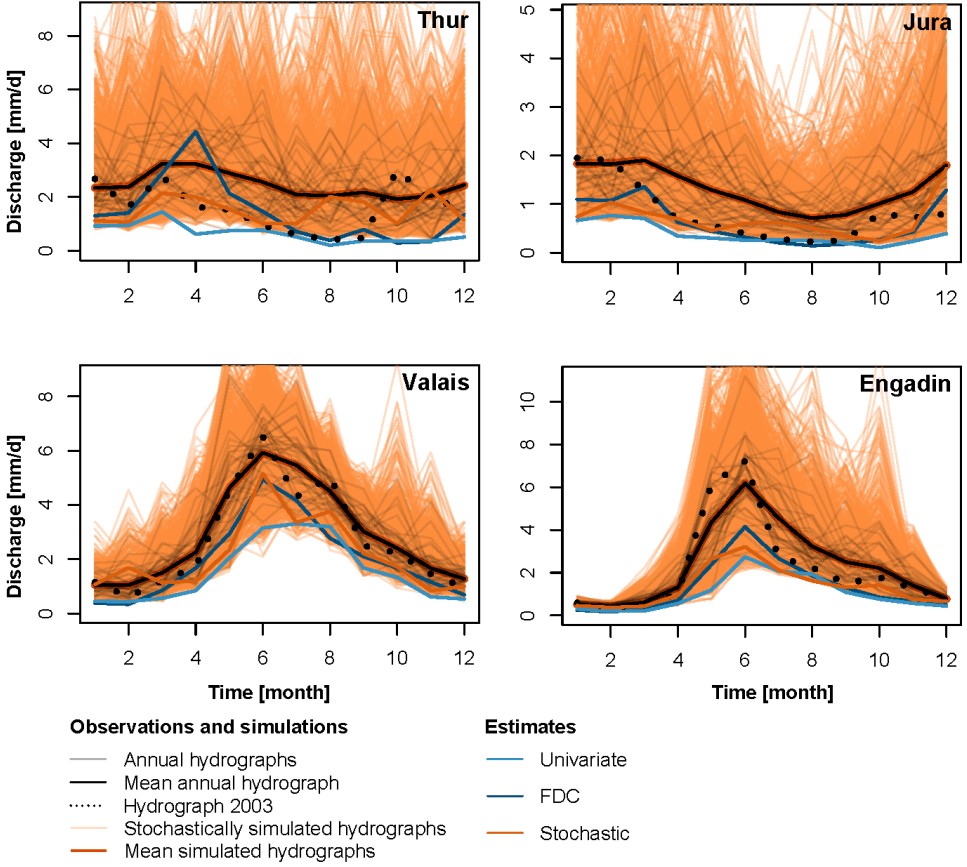

**Figure A1.** Comparison of the 100-year low-flow regime estimates univariate, FDC, and stochastic with stochastically generated hydrographs (orange lines). The observed mean hydrograph (solid line) and the hydrograph of the year 2003 (dotted line) are given in black.

*Author contributions.* The idea and setup for the analyses was developed by MB. MZ did the hydrological model simulations. HZ, MH, and DF provided the future glacier extents. The analyses were performed by MB and discussed with MZ and DF. MB wrote the first draft of the manuscript which was revised by all co-authors and edited by MB.

*Competing interests.* The authors do not have any competing interests





**Table A1.** Summary of 39 climate chains considered: Global circulation model (GCM), regional climate model (RCM), representative concentration pathway (RCP), and grid cell resolution.

| GCM | RCM | RCP | Resolution |
|---|---|---|---|
| ICHEC-EC-EARTH | DMI-HIRHAM5 | 2.6 | EUR-11 |
| ICHEC-EC-EARTH | DMI-HIRHAM5 | 4.5 | EUR-11 |
| ICHEC-EC-EARTH | DMI-HIRHAM5 | 8.5 | EUR-11 |
| ICHEC-EC-EARTH | SMHI-RCA4 | 2.6 | EUR-11 |
| ICHEC-EC-EARTH | SMHI-RCA4 | 4.5 | EUR-11 |
| ICHEC-EC-EARTH | SMHI-RCA4 | 8.5 | EUR-11 |
| MOHC-HadGEM2-ES | SMHI-RCA4 | 4.5 | EUR-11 |
| MOHC-HadGEM2-ES | SMHI-RCA4 | 8.5 | EUR-11 |
| MPI-M-MPI-ESM-LR | SMHI-RCA4 | 4.5 | EUR-11 |
| MPI-M-MPI-ESM-LR | SMHI-RCA4 | 8.5 | EUR-11 |
| ICHEC-EC-EARTH | CLMcom-CCLM5-0-6 | 8.5 | EUR-44 |
| MOHC-HadGEM2-ES | CLMcom-CCLM5-0-6 | 8.5 | EUR-44 |
| MIROC-MIROC5 | CLMcom-CCLM5-0-6 | 8.5 | EUR-44 |
| MPI-M-MPI-ESM-LR | CLMcom-CCLM5-0-6 | 8.5 | EUR-44 |
| ICHEC-EC-EARTH | DMI-HIRHAM5 | 2.6 | EUR-44 |
| ICHEC-EC-EARTH | DMI-HIRHAM5 | 4.5 | EUR-44 |
| ICHEC-EC-EARTH | DMI-HIRHAM5 | 8.5 | EUR-44 |
| ICHEC-EC-EARTH | DNMI-RACMO22E | 4.5 | EUR-44 |
| ICHEC-EC-EARTH | DNMI-RACMO22E | 8.5 | EUR-44 |
| MOHC-HadGEM2-ES | DNMI-RACMO22E | 2.6 | EUR-44 |
| MOHC-HadGEM2-ES | DNMI-RACMO22E | 4.5 | EUR-44 |
| MOHC-HadGEM2-ES | DNMI-RACMO22E | 8.5 | EUR-44 |
| CCma-CanESM2 | SMHI-RCA4 | 4.5 | EUR-44 |
| CCma-CanESM2 | SMHI-RCA4 | 8.5 | EUR-44 |
| ICHEC-EC-EARTH | SMHI-RCA4 | 2.6 | EUR-44 |
| ICHEC-EC-EARTH | SMHI-RCA4 | 4.5 | EUR-44 |
| ICHEC-EC-EARTH | SMHI-RCA4 | 8.5 | EUR-44 |
| MOHC-HadGEM2-ES | SMHI-RCA4 | 2.6 | EUR-44 |
| MOHC-HadGEM2-ES | SMHI-RCA4 | 4.5 | EUR-44 |
| MOHC-HadGEM2-ES | SMHI-RCA4 | 8.5 | EUR-44 |
| MIROC-MIROC5 | SMHI-RCA4 | 2.6 | EUR-44 |
| MIROC-MIROC5 | SMHI-RCA4 | 4.5 | EUR-44 |
| MIROC-MIROC5 | SMHI-RCA4 | 8.5 | EUR-44 |
| MPI-M-MPI-ESM-LR | SMHI-RCA4 | 2.6 | EUR-44 |
| MPI-M-MPI-ESM-LR | SMHI-RCA4 | 4.5 | EUR-44 |
| MPI-M-MPI-ESM-LR | SMHI-RCA4 | 8.5 | EUR-44 |
| NCC-NorESM1-M | SMHI-RCA4 | 2.6 | EUR-44 |
| NCC-NorESM1-M | SMHI-RCA4 | 4.5 | EUR-44 |
| NCC-NorESM1-M | SMHI-RCA4 | 8.5 | EUR-44 |

*Acknowledgements.* We thank the Federal Office for the Environment (FOEN) for funding the project (contract: 15.0003.PJ/Q292-5096). We also thank MeteoSwiss for providing observed meteorological data and the Swiss National Centre for Climate Services (NCCS) for providing the climate model simulations.





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
