# Peer review of "Future shifts in extreme flow regimes in Alpine regions"

_Hydrology and Earth System Sciences, 2019_

## Referee Comment (RC1) · Anonymous Referee #1 · 8 May 2019

General comments (overall quality of the discussion paper)

The authors have presented a nicely prepared study that applies hydrological simulations driven by different meteorological forcings for past and future climate conditions to estimate changes in high- and low-flow conditions for different rain- and melt-dominated catchments in Switzerland. Thereby, different approaches for estimating high- and low-flow regimes are applied and compared, discussing the individual strengths and weaknesses of each approach. A total number of 39 realizations of three different IPCC representative concentration pathways (8 x RCP2.6, 13 x RCP4.5 and 18 x RCP8.5) is compared to reflect the bandwidth of potential change in future discharge conditions in the catchments considered. As the knowledge on potential changes in high- and low-flow conditions is of high societal, economic and ecological importance, the

present study is of high scientific relevance with the thematic focus fitting nicely in the thematic scope of HESS. The formal requirements for publishing are almost fully met as reflected in the low number of technical corrections suggested in the following. My major concerns relate to the comparisons of current and future discharge conditions. While the hydrological simulations representing the data basis for the estimation of current flow conditions are driven by meteorological observations, the simulations for future discharge conditions are based on climate conditions simulated by different GCM-RCM combinations. Although the downscaling approach of quantile mapping has been applied to statistically correct biases in the climate simulations for past and future conditions, the meteorological conditions in the observational data set applied for the hydrological simulations for the past are still not identical to the RCM simulated climate simulations for the past. This induces biases when it comes to the comparison of past and future discharge conditions calculated on the basis of meteorological data from the two different origins. Differences in the meteorological data sets can be expected to occur at shorter time scales (as highly relevant for extreme weather and hydrological conditions) as well as with respect to the interdependency of various meteorological variables, e.g. temperature, humidity and precipitation (such inter-variable consistency is not conserved applying quantile mapping). The existing differences in the meteorological observations and climate simulations with their respective hydrological effects for the past become clearly evident looking at the differences between the hydrological simulations achieved using meteorological observations and simulations in the case of both mean and extreme conditions in Figure 6. Moreover, calibration of the hydrological model seems to be carried out using meteorological observations for the past. While less important compared to the issue described before, calibration can indirectly compensate deficiencies in the meteorological input applied during calibration. Applying the same calibrated parameter set using different meteorological input (with very likely different deficiencies, e.g. those related to the quantification of precipitation) might lead to inconsistencies in the model results. A way to avoid the inconsistencies arising from different meteorological input for the past and future would

be to correct the RCM simulations for the past and future statistically (as done in the present study) and later only compare the differences between the hydrological simulations for the past and the future using past and future climate simulations as input for the hydrological simulations. Figure 6 follows this direction but all other results seem to compare current and future conditions generated on the basis of different meteorological input (station observations vs RCM simulations). Using the just outlined approach the hydrological simulations would be better comparable as (hopefully) the same systematic biases are found in the simulations for the past and future. As a large number of RCM simulations are providing meteorological conditions for the past in this study, this would require defining some sort of "one hydrological reference" (e.g., the multi-model mean) the different scenario simulations can then be compared to. Apart from this main point of criticism, some further points for improvement remain, which are described in the "Specific comments" and mostly represent suggestions for clarification as well as options to make the contents of the study easier transportable to the reader. A final issue to be mentioned is, that at some point a closer linkage of the hydrological results to the climate change signal in the applied climate scenarios for Switzerland (e.g., by showing and discussing the temperature and precipitation change) would be beneficial to interpret the presented hydrological changes. However, considering the large number of scenarios and realizations this is probably beyond the scope of the article and should hence not be an issue that needs to be addressed in the revision.

As a final recommendation, I suggest to accept the article for publication in HESS given all issues have been addressed properly (major revisions)

Specific comments (individual scientific questions/issues)

1) p.3, l.24: The authors describe existing approaches for the generation of discharge time series and then name the approach chosen in their study. Although more detailed information is provided in the "Methods section", it might be valuable for the reader to receive some brief description, in how far the applied approach differs from or matches the ones described shortly before. Maybe the authors could add some information on

this here?

2) p.4, Figure 1: This figure is considered very important for the reader to get an overview of the study regions. However, apart from the outlines of the study regions, the map seems rather general and would benefit from some additional detail. Maybe it would be possible to support orientation for non-EU readers e.g., by modifying Figure 1, including a larger scale overview map linked to the original map as well as by adding some more details e.g., the names of larger rivers, lakes or cities?

3) p.4, l.7: The authors include the information "second step" in brackets - this approach helps to structure the workflow and might be a good extension also for the first and following steps, which are currently referred to as part of the describing text. Having all the steps in brackets would make it easier to navigate through the workflow for the reader.

4) p.4, l.9: The discharge series used for the estimation of extreme regimes are based on hydrological simulations using observed meteorological conditions for past discharge conditions and simulated meteorological conditions (climate model output) for future discharge conditions. Given the biases in currently available climate simulations the reader here wonders if some correction has been applied before the application of the climate simulations for past and future in this study. Although this is later explained adding "bias-corrected" and "downscaled" might satisfy the readers curiosity at this early point.

5) p. 5, Figure 2: Number 1 in the workflow (comparison procedures) leads to some confusion from my point of view. I understand from the workflow description starting on p.4, l.5 that the different methods for estimating extreme flow regimes were tested, however this comparison seems to require the hydrological simulations of the left side of Figure 2 as input already, which somehow conflicts with the rank in the overall workflow. Some further confusion in Figure 2 from my point of view arises in the context of the "Data" column. The caption says "A" introduces the simulated data used, however
the two plots rather seem to show the meteorological input applied (climate simulations and observations) for the discharge simulations. On the other hand, the data flow from "A" to "C" (the "Estimation" box), as well as that from "A" to "B" requires the two data inputs from "A" to be the discharge simulations achieved on the basis of the two different meteorological data sources (climate simulations and observations). Maybe the authors could clarify these issues in an updated version of Figure 2. It thereby would be an option to use different colors for meteorological input and hydrological model results - while white and orange are used in the figure, the caption does not clarify whether the colors are used in this sense. Maybe it would also be beneficial to use different signatures/colors for the current and future estimates in box "C" and more clearly link them to the discharge simulations based on meteorological observations and climate simulations following the argumentation on p.4, l.9? Maybe also the PREVAH model, which is later mentioned to be used for the hydrological simulations, should be already integrated in the figure to complete the workflow?

6) p.5, l.12: The authors describe that validation was performed for the period 1983-2005, presumably using meteorological observations as input for the PREVAH model. As climate simulations are used for the model runs for future conditions, I wonder if any validation using climate simulations for the past has been carried out beside the results shown in Figure 6 (e.g., by using hindcast simulations that reproduce weather conditions and allow a comparison of simulated and observed discharge at daily basis)?

7) p.5, l.12: The authors explain that the applied hydrological model has been calibrated for the period 1993-1997. Assuming that the model was driven by meteorological observations while calibrating, the question of systematic biases due to calibration under observed meteorological conditions as well as the application of different meteorological input for the past and future arises. From my perspective both the application of different meteorological input for the past and future (meteorological stations vs climate model simulations) as well as the fact that calibration has been carried out for past observed climate conditions, whereas the simulations are presumably carried out

with the same parameter set for future simulated climate conditions, need some discussion. It might be the case that calibration compensates for some of the biases in the observed meteorological input data (e.g. an undercatch in precipitation), which are not present or at least different in the case of meteorological simulations. Particularly in snow dominated regions differences in temperature and precipitation between the two data sources can lead to different water storage in the snow pack affecting the simulated discharge conditions. A comparison of the statistical discharge characteristics achieved for the past using meteorological observations (for which the model was calibrated) to those achieved for the past using climate simulations (which were not used in the calibration) over a climatological period of time could show the differences in the discharge characteristics (e.g., using flow duration curves), which might somehow also affect the comparability of past and future discharge simulations in the present study. Maybe the authors could add some discussion on these issues to the manuscript.

8) p.6, l.2: Two glacier models are applied depending on the length of the glacier considered. It would be interesting why the extended GloGEMflow model is not applied to smaller glaciers (I understand that glacier flow is less important in this case but are there any reasons applying only the GloGEMflow model for all lengths is inadequate?) and what the differences between the results of both models would be - the latter might give an impression of the impact on the overall results induced by using the two different glacier models.

9) p.6, l.11: Does radiation refer to shortwave or longwave radiation or both? Please clarify.

10) p.6, l.11 and l.17: Has the approach of Quantile Mapping been applied for all meteorological variables listed in l. 11? Particularly short- and longwave radiation recordings are often not available far back in the past, reducing the possibility to statistically correct the simulations for these variables. Not correcting all variables used as input for the hydrological model would explain differences between the hydrological model results achieved using meteorological observations and those based on meteorological

simulations.

11) p.6., l.18: Does "model chains" here refer to "GCM-RCM combinations" - if yes, this term would be more precise and should be used alternatively. Or are "GCM-RCM combinations for different scenarios and realizations" the content behind the "model chains" - in any case, please be more precise and avoid the use of "model chains" as it can include quite a lot of pre- and postprocessing steps when used without further clarification.

12) p.6, l.19: Please rephrase to "for the locations of various meteorological stations".

13) p.6, l.23: Please add "for topographic corrections" or a similar completion to "was additionally used".

14) p.6, l.34: Although later specified, it is not clear whether the applied discharge series is that observed for the gauging stations or the hydrological simulations - maybe adding "(here the simulated discharge for past and future conditions)" would make this clear as early as possible.

15) p.6, eq. 1: What does "i" stand for?

16) p.6. l.14: "Model chains" is not really precise (see comment 11), please provide additional information.

17) p. 7, l.24: Here the reader wonders how these unrealistic estimates are handled and how they affect the robustness of the study findings - would it be an option to already point out here that the univariate technique will not find further consideration in the study as later mentioned (see p. 11, l. 2)?

18) p. 7, l.3: Does this mean the typical flow regime assumed is allowed to be different for past and future conditions depending on the simulated discharge time series - this is an important point and needs to be included.

19) p. 8, l. 6: Is the number necessarily as high as 1500 years to apply this approach

or is it 1500 years because the authors carried out simulations for 1500 years (see p.7, l. 11)? Maybe adding "here" before 1500 would avoid any speculation.

20) p.8, l. 13: The authors describe that a "control regime" was generated on the basis of discharge simulations achieved using meteorological observations as input for the hydrological model and that a number of "reference regimes" have been derived from the hydrological simulations based on the different GCM-RCM combinations, resulting in a range of current regime estimates. This is not fully in line with previous statements that describe that the discharge simulations representing past discharge conditions are based on meteorological observations (see p.4, l. 9) so it would be beneficial to modify the statements at p.4, l. 9 accordingly.

21) p.8, Figure 3: Please add to the caption that this figure is just a schematic illustration of the comparison approach to make clear the underlying data is not generated in this study and should not be interpreted.

22) Figure 4 & 5: For me it is not quite clear whether the extreme regime estimates for this control setting are derived based on simulated discharge (see p.8, l.13) or on the discharge observations also illustrated in Figure 4 and 5. However this makes a difference, as using discharge observations for the extreme regime estimation would represent a "perfect setup" to only quantify the uncertainties in the estimation approaches, while using discharge simulations based on meteorological observations (previously defined as "control conditions") would rather illustrate the uncertainty in the whole control setting (which according to the described study design includes the uncertainties from the hydrological modelling using meteorological observations as input) as well as the uncertainties from the individual regime estimation method. Ideally, the extreme regime estimation methods would also be tested using the reference simulations (achieved using climate model data for the past as input for the hydrological model) to show the uncertainty additionally induced by applying climate simulations as hydrological model input (as done for low-flow conditions in Figure 6 using only the FDC method).

23) p. 11, Figure 6: This figure is nice as it for the first and only time compares the results achieved for meteorological observations (control simulations) and the climate simulations - is there any reason this is only done for low-flow conditions? There also is some room for improvement with respect to the graphical realization: The different signatures show the "control simulations" (observed meteorological input for the hydrological model) and "reference simulations" (data from different GCM-RCM combinations as input for the hydrological model). The description "Control simulation" and "Climate simulations" is a little confusing, as for the control simulations the word simulations represents hydrological simulations, while in the case of the climate simulations the word simulations refers to the meteorological input applied. I therefore suggest denoting the signatures "control simulations" and "reference simulations" with a detailed description (as already provided) in the figure caption. Moreover, I would suggest to choose a dashed-type of signature for the upper and lower borders of the reference simulations (climate simulations) similar to the future range in Figure 7) to make them distinguishable from the control simulations.

24) p. 11, l.5: I would in general avoid the word "model chain" and replace it with something more precise (e.g. GCM-RCM combination). "Model chain" is a rather wide term that can include a hydrological model, the extreme regime estimation or the downscaling and bias-correction procedure (see also comment 11).

25) p. 11, l. 8: Better replace "observed data" with "meteorological observations" to be more precise. I would also replace "means" with "suggests" as it is rather a hint in that direction and not a fact (also the line is in the spread range, the individual reference simulations can be quite far from the line).

26) p. 11, l. 9: The authors explain the overestimation in summer with an overestimation in RCM-simulated precipitation. This is plausible, but shouldn't the precipitation statistics in the RCM data match that of the observations after application of the quantile mapping approach? Maybe the authors could deepen the discussion on this issue in the updated version of the manuscript.

[Figure]

27) p. 12, Figure 7: I assume that the current regime is based on the hydrological simulations achieved using meteorological observations as input for the hydrological model (as described earlier in the text, see p.4, l.9). I see a certain weakness of this study in directly comparing the hydrological simulations achieved with meteorological observations (current) and climate simulations (future). While the applied bias correction fits the statistics of the climate simulations for the past to those of the observations (but also only in case of those meteorological variables that are bias corrected, see comment 10), the data sets still are not identical for the past and can be expected to induce different hydrological reactions, particularly when it comes to extreme events. I think the study would have benefitted from using a control regime that is also based on climate simulations, in this case for the past (e.g., multi-model mean of all hydrological model results achieved using different GCM-RCM combinations). Otherwise the systematic differences in the driving data make the results hard to compare. This at least needs to be discussed in the updated manuscript.

28) p.12, Figure 7: The solid (current) and dashed lines (surrounding the shaded areas) in the case of the "mean conditions" are black in the legend but seem to be grey in the charts - this should be corrected. The caption indicates that the "normal" regimes are provided as a reference in grey, but are these the same as the "mean conditions", if yes please try to avoid using different words for the same content to make the figure easier to understand. The legend should in this case also be modified to link a grey solid line to the current state and grey dashed lines to the boundaries surrounding the grey areas as described in the caption. Finally, the areas where orange and light-blue areas overlap are extremely hard to distinguish from light-grey areas making the plots extremely hard to read. Please try to update the plots to make them easier to read - maybe there is simply too much information in them? Separating mean and extreme conditions would already make a big difference. Moreover, rather or additionally discussing the multi-model mean instead of the spread in future discharge achieved from all GCM-RCM simulations could reduce the displayed information for the sake of readability and interpretability.

29) p.12, l.3: I would suggest to rephrase "These chains" to "Here, the different realizations".

30) p.12, l.3: There seems to be also a distinct reduction in mean and extreme conditions for FDC in both rain dominated catchments in spring. Apart from RCP4.5 in the Jura catchment this reduction is clearly evident in both rain dominated catchments and all scenarios. Maybe this could also be an issue of discussion.

31) p.12, l.8: Replace "differences in" with "differences between".

32) p.12, l. 9: Why has RCP4.5 been excluded from Figure 8. If it is because you expect the changes to be in between those resulting from applying RCP2.6 and RCP8.5 please add a short sentence including this information to the text.

33) p.13, l. 1: Do you mean "different" instead of "distinct" - "different" would be better from my point of view as you explain how the changes differ in the following lines.

34) p.14, Figure 8: "Mean" is used in the plot and "normal" is used as a describing term in the caption - I would suggest using one of the terms as the counterpart to "extreme" consistently in the figures and throughout the manuscript. Please rephrase "the second three rows" to "the last two rows" in the caption as there are only five rows in total.

35) p.14, l.2: I would suggest to include a brief description of the main findings from Figure 9 before moving on to Figure 10.

36) p.14, l.3: Are the changes really "independent of the estimation technique used (FDC/stochastic)" or just in a similar order of magnitude or comparable?

37) p.15, Figure 9: As Figure 7 and Figure 9 share many characteristics, the options for improvement described for Figure 7 (comment 28) mostly also apply to Figure 9. Moreover, the y-axes in the case of the Thur catchment need adjustment towards higher maximum y-values as the maxima are cut out in the case of all scenarios.

38) p.16, Figure 10: Please rephrase "the second three rows" with "the last two rows"

in the caption as there are only five rows in total.

39) p.17, l.17: Better change "include future climate scenarios" to "include the assumptions underlying the applied future global climate scenarios". Maybe also change "the choice of a hydrological model" to "the uncertainties inherent in the hydrological model results".

40) p.17, l.33: Please replace "regions but the low-flow" with "regions but with the low-flow".

Technical corrections (typing errors, etc.)

1) p.2, l.6: Better rephrase to "... in the future ...".

2) p.5, l.9: I think "glacier melt" in two separate words is the most commonly used form.

3) p.18, l.23: Please replace "which coincides" with "which coincide" and "high flow" with "high-flow".

---

## Referee Comment (RC2) · Anonymous Referee #2 · 16 May 2019

Comments on: Future shifts in extreme flow regimes in Alpine regions. Brunner, M.I. et al. Hydro. Earth Syst. Sci. Discuss, https://doi.org/10.5194/hess-2019-144

Overview: This paper considers future changes in 'extreme flow regimes' in 19 hydrological regions in Switzerland as interpreted from annual flow duration curves and annual hydrographs aggregated as monthly average flows. Two novel methods are applied to generate estimates for the 100-year regimes for the current climate (using simulations driven by observed meteorological data) and a future climate (with simulations driven by bias-adjusted climate model output). The methods are illustrated using four example regions, representing both rainfall-dominated and snowmelt-dominated flow regimes, before being applied to the 19 hydrological regions. The results point towards patterns of change which are distinct for rainfall-dominated vs. snowmelt-dominated

flow regimes, and which are consistent with previous work and with changes in the hydrometeorological drivers. The authors also propose that the two approaches applied give similar and consistent results.

General comments: Overall, I find the quality of the work presented to be quite high. The topic addressed has important practical applications, and the introduction and application of alternative methods for interpreting changes in extreme conditions from data series of limited length (30-year) is much needed in climate change impacts research. The manuscript is well-written, has a structure and figures that are in most cases clear and easy to follow. I am, however, in agreement with Reviewer #1 that the major weakness of the work is the comparisons made between simulations based on observations (for the current climate) with those based on climate model data (for the future climate) in order to interpret expected changes in flow regimes. One can only justify comparisons between reference and future simulations based on climate model data, due to the inevitable discrepancies between simulations based on observed data vs. climate model data. This needs to be corrected before the manuscript can be considered for publication.

Secondly, the choice of the use of a direct stochastic simulation method rather than an 'indirect' stochastic method (which is mentioned but not discussed at all) should be presented in more detail. In particular, the use of a direct stochastic simulation of discharge from 'sampled' discharge (either simulated or observed) entails the assumption that events with long return periods come from the same population as those with shorter return periods (and can therefore be extended based on their power spectrum or using other extrapolation techniques). With 'indirect' methods, this constraint is less severe, in that driving factors producing the flows (e.g. precipitation and initial conditions) can be modelled individually and thus will potentially produce a wider range of likely flow conditions and as of yet unobserved combinations. There are nevertheless many drawbacks with the 'indirect' methods, so the authors should use the opportunity to here to highlight why they have chosen their particular stochastic approach.

My final reservation about the manuscript as it is currently presented is that the discussion of the performance of FDC and stochastic methods is limited and rather superficial. I don't see that the two methods give uniformly similar results, particularly for high flow regimes in rainfall-dominated catchments. The magnitude of the change in maximum discharge under a future climate are both higher and lower for the stochastic method than for the FDC method and the projections give different seasons for the maximum. In addition, results comparing simulations based on observed vs. climate data for the current climate are not shown for high flows. Given discrepancies between the two methods found in other figures, one would also like to be able to assess the correspondence between the simulations for high flows. A full development and discussion of the results, with reference to the different aspects considered here (i.e. snowmelt-vs. rainfall-dominated catchments and high flow vs. low flow regimes) would significantly enhance the contribution of this work to the scientific literature. Overall, I find this to be a valuable piece of work will be worthy of publication, once the issues raised above have been addressed. Otherwise, I have only a few additional minor comments, questions and proposed corrections as given below.

Specific comments:

P1 – Abstract: Well-presented, but at this point in the manuscript I also needed a clarification as to what is meant by 'flow regime'. Perhaps you should more clearly emphasis that in your end results you are interpreting/analysing 'flow regimes' using annual hydrographs comprised of monthly averaged flows. Particularly, for the case of high flow regimes, the time unit analysed is of interest.

P2 L24-25 I don't agree with the statement that the focus on an individual characteristic "neglects the pre-conditions of low- and high-flow events", at least in terms of how most climate impact studies consider changes in this variables. For example, if one is evaluating changes in maximum daily flows based on daily simulations of 30-year periods for a present and a future climate, potentially higher flows in the period preceding the largest events are also represented in those simulations, i.e. they are not

neglected. It is true that by considering an index which targets a broader time window or discharge range you are able to more directly interpret why these changes occur, e.g. because discharge is elevated for a longer time period and is thus more susceptible to extreme precipitation. In the 'standard' one characteristic approach, this would require a further analysis. A similar argument can be made for low flow indices, i.e the antecedent conditions are simulated and can be analysed if one requires additional information as to factors responsible for the estimated changes. However, to simply state that pre-conditions are neglected is misleading and incorrect.

P3 L15-20 As mentioned in the general comments, this paragraph needs to justify why direct stochastic simulation of discharge is used in this work. I would therefore suggest that the paragraph opens with a sentence describing what stochastic simulation is in general, and that this is followed by a more thorough discussion of the advantages and disadvantages of indirect vs. direct approaches, before you jump into a detailed discussion of options for direct simulation. It is important that you place the direct stochastic simulation method you have chosen within its broader context, and argue for why it is preferable and suitable for use here. In particular, there is a wide and growing literature on 'indirect' stochastic simulation which should be covered here with at least 2-3 sentences.

P6 L17 'regional downscaling approach based on quantile mapping'….more detail is needed. In particular, it is not clear in the description given in L20-L21 at what point the bias correction with quantile mapping is applied. I assume that the same data which is used for the simulations based on observed data is used for the bias correction….is this correct? In addition, what is the time period of the observed data used for bias correction?

P6 L18 ….'39 model chains (Table A1'. The choice of models used should be discussed, rather than leaving it to the reader to decipher a table in the appendix in order to determine how many different GCMs vs. RCMs vs. RCPs vs. grid resolutions are represented by the 39 simulations. The different models, etc. represented in the en-

semble will indeed have an effect on both the mean values estimated and the spread about the mean, and the 39 model chains used here are only a subset of available EUROCORDEX simulations. In particular, it is unclear to me why both EUR-11 and EUR-44 grid resolutions have been used, if the climate model data are used for hydrological simulations at a 200-m grid resolution. I suspect that the EUR-44 simulations have been included to give a larger ensemble; however, this also means that the GCM-RCM combinations that are available for both EUR-11 and EUR-44 are more heavily weighted in the ensemble. A brief (2-3 lines) discussion of the composition of the ensemble is therefore needed in the manuscript.

P7 L13-14 Need to also generate stochastic series for the current 'conditions' using the hydrological model simulations based on the 39 model chains for the period 1981-2010.

P8 L2 – 'Long-term mean'. . .can be more specific, i.e. the mean regime over the period 1981-2010? I assume from your figures that this is aggregated by month?

P8 L14-15 It is this 'reference' simulation which should actually be the 'control' simulation and be further used to evaluate future changes

P8 L17 'were treated separately'. . ..not sure what is meant by this. Perhaps simply that the results were grouped by RCP?

P10 L5-7 Why is the seasonality so variable between the FDC, Stochastic and Univariate estimates in the rainfall-dominated Thur and Jura catchments? Doesn't this undermine the credibility of these methods for considering high flows in these and other rainfall-dominated catchments?

P10 L10-12 Given my comment above, I am also curious as to what Fig. A1 looks like for high-flow regime estimates? In particular, the stochastic results for Jura in Fig. 5 suggest that the method doesn't perform better than the univariate approach in that case.

P11 Sec 3 Would also like to see a comparison of the climate simulations and the control simulation for high flows, similar to that shown for Fig 6 for low flow regimes.

P12 L10-11: Changes in mean flow of up to 50%, etc. These estimates are not reliable because you are comparing the results of simulations based on climate model data (future) with those based on observations (current), and thus are also including some the error illustrated in Figure 6 in your estimates.

P17 Section 4.1: In addition to discussing the overall merits of the two methods, it would also be useful to see a discussion of their relative performance for high vs. low flow regimes and for rainfall-dominated vs. snowmelt-dominated catchments. I also find that the second paragraph in this section is too general and should focus on discussing and expanding the results you have presented in more depth. It would also be useful, for example, if you could highlight aspects of the methods you have used which are better for quantifying changes in flow regimes useful for management purposes, relative to those commonly used for climate impact studies.

P18 L15-16: 'Both were found to provide realistic, mutually-agreeing results'. I think that this is bit overstated. In particular, the results for the high flow regimes for the 19 regions (Fig. 10) show a similar direction of change (in most cases), but the magnitude is in some cases much higher with the Stochastic method, and this would have significant implications for their application in practice. Technical comments:

P1 – Keypoints: (L22) – 'are changing' should be 'will change', i.e. you have not examined patterns of change under current conditions, which is what the English used here implies

P2 L17-18: Considering rephrasing, for example to: For planning purposes and river basin management, however, estimates not only for normal conditions but also for extreme conditions are needed.

P4 L3. . .should add 'under the current climate' after 'regime.'

P12 L4 Replace 'expressed' with 'pronounced'

P14 Fig 8; P16 Fig10 caption: In line 3 should be 'The top three rows show relative changes, and the bottom two rows show changes in months'

P18 L8-9: Replace 'We here showed' with 'We have shown here'
* * *

---

## Author Comment (AC1) · 11 Jun 2019

**Reviewer 1**

**General comments (overall quality of the discussion paper)**

The authors have presented a nicely prepared study that applies hydrological simulations driven by different meteorological forcings for past and future climate conditions to estimate changes in high- and low-flow conditions for different rain- and melt-dominated catchments in Switzerland. Thereby, different approaches for estimating high- and lowflow regimes are applied and compared, discussing the individual strengths and weaknesses of each approach. A total number of 39 realizations of three different IPCC representative concentration pathways (8 x RCP2.6, 13 x RCP4.5 and 18 x RCP8.5) is compared to reflect the bandwidth of potential change in future discharge conditions in the catchments considered. As the knowledge on potential changes in highland low-flow conditions is of high societal, economic and ecological importance, the present study is of high scientific relevance with the thematic focus fitting nicely in the thematic scope of HESS. The formal requirements for publishing are almost fully met as reflected in the low number of technical corrections suggested in the following.

My major concerns relate to the comparisons of current and future discharge conditions. While the hydrological simulations representing the data basis for the estimation of current flow conditions are driven by meteorological observations, the simulations for future discharge conditions are based on climate conditions simulated by different GCM-RCM combinations. Although the downscaling approach of quantile mapping has been applied to statistically correct biases in the climate simulations for past and future conditions, the meteorological conditions in the observational data set applied for the hydrological simulations for the past are still not identical to the RCM simulated climate simulations for the past. This induces biases when it comes to the comparison of past and future discharge conditions calculated on the basis of meteorological data from the two different origins. Differences in the meteorological data sets can be expected to occur at shorter time scales (as highly relevant for extreme weather and hydrological conditions) as well as with respect to the interdependency of various meteorological variables, e.g. temperature, humidity and precipitation (such inter-variable consistency is not conserved applying quantile mapping). The existing differences in the meteorological observations and climate simulations with their respective hydrological effects for the past become clearly evident looking at the differences between the hydrological simulations achieved using meteorological observations and simulations in the case of both mean and extreme conditions in Figure 6. Moreover, calibration of the hydrological model seems to be carried out using meteorological observations for the past. While less important compared to the issue described before, calibration can indirectly compensate deficiencies in the meteorological input applied during calibration. Applying the same calibrated parameter set using different meteorological input (with very likely different deficiencies, e.g. those related to the quantification of precipitation) might lead to inconsistencies in the model results. A way to avoid the inconsistencies arising from different meteorological input for the past and future

would be to correct the RCM simulations for the past and future statistically (as done in the present study) and later only compare the differences between the hydrological simulations for the past and the future using past and future climate simulations as input for the hydrological simulations. Figure 6 follows this direction but all other results seem to compare current and future conditions generated on the basis of different meteorological input (station observations vs RCM simulations). Using the just outlined approach the hydrological simulations would be better comparable as (hopefully) the same systematic biases are found in the simulations for the past and future. As a large number of RCM simulations are providing meteorological conditions for the past in this study, this would require defining some sort of "one hydrological reference" (e.g., the multimodel mean) the different scenario simulations can then be compared to. Apart from this main point of criticism, some further points for improvement remain, which are described in the "Specific comments" and mostly represent suggestions for clarification as well as options to make the contents of the study easier transportable to the reader.

A final issue to be mentioned is, that at some point a closer linkage of the hydrological results to the climate change signal in the applied climate scenarios for Switzerland (e.g., by showing and discussing the temperature and precipitation change) would be beneficial to interpret the presented hydrological changes. However, considering the large number of scenarios and realizations this is probably beyond the scope of the article and should hence not be an issue that needs to be addressed in the revision.

As a final recommendation, I suggest to accept the article for publication in HESS given all issues have been addressed properly (major revisions)

**Reply:** *Thank you for your suggestion of replacing the regimes derived from a control run (observed meteorology) by regimes derived from a reference run (simulated meteorology). We will follow your suggestion and replace the regimes derived from the control run by a multi-model mean computed from the 39 reference runs. All the figures affected will be updated accordingly.*

**Specific comments (individual scientific questions/issues)**

1) p.3, l.24: The authors describe existing approaches for the generation of discharge time series and then name the approach chosen in their study. Although more detailed information is provided in the "Methods section", it might be valuable for the reader to receive some brief description, in how far the applied approach differs from or matches the ones described shortly before. Maybe the authors could add some information on this here?
**Reply:** *We will add a brief description of the differences of the simulation approach employed here as compared to existing approaches. As opposed to classical phase randomization approaches, this approach does not rely on the empirical distribution but uses the flexible, four-parameter kappa distribution (Hosking, 1994), which allows for the generation of a wide range of realizations of high and low discharge values.*

2) p.4, Figure 1: This figure is considered very important for the reader to get an overview of the study regions. However, apart from the outlines of the study regions, the map seems rather general and would benefit from some additional detail. Maybe it would be possible to support orientation for non-EU readers e.g., by modifying Figure 1, including a larger scale overview map linked to the original map as well as by adding some more details e.g., the names of larger rivers, lakes or cities?

**Reply:** *We will add an inlet map of Europe, showing the location of Switzerland within Europe. Since the map is already quite busy we would prefer to restrict the labels to the ones already present.*

3) p.4, l.7: The authors include the information "second step" in brackets - this approach helps to structure the workflow and might be a good extension also for the first and following steps, which are currently referred to as part of the describing text. Having all the steps in brackets would make it easier to navigate through the workflow for the reader.

**Reply:** *We will add the number of the step in brackets.*

4) p.4, l.9: The discharge series used for the estimation of extreme regimes are based on hydrological simulations using observed meteorological conditions for past discharge conditions and simulated meteorological conditions (climate model output) for future discharge conditions. Given the biases in currently available climate simulations the reader here wonders if some correction has been applied before the application of the climate simulations for past and future in this study. Although this is later explained adding "bias-corrected" and "downscaled" might satisfy the readers curiosity at this early point.

**Reply:** *The information will be added to the text.*

5) p. 5, Figure 2: Number 1 in the workflow (comparison procedures) leads to some confusion from my point of view. I understand from the workflow description starting on p.4, l.5 that the different methods for estimating extreme flow regimes were tested, however this comparison seems to require the hydrological simulations of the left side of Figure 2 as input already, which somehow conflicts with the rank in the overall workflow.

Some further confusion in Figure 2 from my point of view arises in the context of the "Data" column. The caption says "A" introduces the simulated data used, however the two plots rather seem to show the meteorological input applied (climate simulations and observations) for the discharge simulations. On the other hand, the data flow from "A" to "C" (the "Estimation" box), as well as that from "A" to "B" requires the two data inputs from "A" to be the discharge simulations achieved on the basis of the two different meteorological data sources (climate simulations and observations). Maybe the authors could clarify these issues in an updated version of Figure 2. It thereby would be an option to use different colors for meteorological input and hydrological model results - while white and orange are used in the figure, the caption does not clarify whether the colors are used in this sense. Maybe it would also be beneficial to use different signatures/colors for the current and future estimates in box "C" and more clearly link them to the discharge

simulations based on meteorological observations and climate simulations following the argumentation on p.4, l.9? Maybe also the PREVAH model, which is later mentioned to be used for the hydrological simulations, should be already integrated in the figure to complete the workflow?

**Reply:** *Thank you for your suggestions. Figure 2 is divided into two parts. The "Data" part and the "Estimates" part. We consider the model simulations to be part of the data generation since the main focus and innovation of our manuscript lies on the "Estimation" part. We did therefore not include the hydrological model into the figure. When we talk about data in this figure, we refer to discharge data, be it observed or simulated. This will be specified to avoid confusion.*

6) p.5, l.12: The authors describe that validation was performed for the period 1983-2005, presumably using meteorological observations as input for the PREVAH model. As climate simulations are used for the model runs for future conditions, I wonder if any validation using climate simulations for the past has been carried out beside the results shown in Figure 6 (e.g., by using hindcast simulations that reproduce weather conditions and allow a comparison of simulated and observed discharge at daily basis)?

**Reply:** *The hydrological model was indeed validated using meteorological observations as input for the PREVAH model (see p. 5, l:12-14). As suggested in one of the next comments, we compared the FDCs for the reference period 1981-2010 derived based on observed meteorology to FDCs derived for the same period using the simulated meteorological data generated by the 39 GCM-RCM combinations. This analysis showed that the FDCs are reproduced well in most catchments, except for the catchment Engadin. As shown in Figure 6, the model was also validated with respect to the reproduction of past low- and high-flow regimes as these were the focus of this study. We will extend Figure 6 by including the high-flow results.*

7) p.5, l.12: The authors explain that the applied hydrological model has been calibrated for the period 1993-1997. Assuming that the model was driven by meteorological observations while calibrating, the question of systematic biases due to calibration under observed meteorological conditions as well as the application of different meteorological input for the past and future arises. From my perspective both the application of different meteorological input for the past and future (meteorological stations vs climate model simulations) as well as the fact that calibration has been carried out for past observed climate conditions, whereas the simulations are presumably carried out with the same parameter set for future simulated climate conditions, need some discussion.

**Reply:** *It has been confirmed by Krysanova et al. (2018) who did a review study on the performance of hydrological models under climate change that a good performance of hydrological models in the historical period increases confidence in projected impacts under climate change. We found that the hydrological model well reproduces the hydrological regimes analyzed in this study in the reference period and therefore assume that it will also reliably simulate hydrological regimes under future conditions. A short discussion about this issue will be added to the manuscript.*

It might be the case that calibration compensates for some of the biases in the observed meteorological input data (e.g. an undercatch in precipitation), which are not present or at least different in the case of meteorological simulations. Particularly in snow dominated regions differences in temperature and precipitation between the two data sources can lead to different water storage in the snow pack affecting the simulated discharge conditions. A comparison of the statistical discharge characteristics achieved for the past using meteorological observations (for which the model was calibrated) to those achieved for the past using climate simulations (which were not used in the calibration) over a climatological period of time could show the differences in the discharge characteristics (e.g., using flow duration curves), which might somehow also affect the comparability of past and future discharge simulations in the present study. Maybe the authors could add some discussion on these issues to the manuscript.

**Reply:** *As shown in Figure 6 in the manuscript, the water balance of most catchments is well represented by using the simulated meteorological data instead of the observed ones. We followed your suggestion and did the validation also on the FDCs directly (Figure 1). The results show that the FDCs are mostly well reproduced by using the simulated meteorological data to simulate discharge. An exception is, as in Figure 6, the Engadin, where high flows are slightly overrepresented. We will discuss in the manuscript that this overestimation might be related to the univariate bias correction applied which might not perfectly reflect the interplay between temperature and precipitation and therefore the timing of snowmelt processes* (Meyer et al., 2019)*.*

[Figure]

**Figure 1: FDCs derived from the control run over the period 1981-2010 (black line) and the 39 reference simulations for the same period (grey lines) for the four catchments Thur, Jura, Valais, and Engadin.**

8) p.6, l.2: Two glacier models are applied depending on the length of the glacier considered. It would be interesting why the extended GloGEMflow model is not applied to smaller glaciers (I understand that glacier flow is less important in this case but are there any reasons applying only the GloGEMflow model for all lengths is inadequate?) and what the differences between the results of both models would be - the latter might give an impression of the impact on the overall results induced by using the two different glacier models.

**Reply:** *For small glaciers a simplified glacier model is indeed used (vs. ice-dynamic model for glaciers >1 km). As shown in the original publication describing the glacier model* (Zekollari et al., 2019)*, for small glaciers the difference between the modeled glacier evolution with the simplified model and the ice-dynamic model is very small though - the difference is found to increase with increasing glacier elevation range (see Figure 12 in* Zekollari et al., 2019*).*

9) p.6, l.11: Does radiation refer to shortwave or longwave radiation or both? Please clarify.
**Reply:** *We will clarify that radiation refers to shortwave radiation.*

10) p.6, l.11 and l.17: Has the approach of Quantile Mapping been applied for all meteorological variables listed in l. 11? Particularly short- and longwave radiation recordings are often not available far back in the past, reducing the possibility to statistically correct the simulations for these variables. Not correcting all variables used as input for the hydrological model would explain differences between the hydrological model results achieved using meteorological observations and those based on meteorological simulations.
**Reply:** *The quantile mapping approach was applied for all the variables. This will be specified in the text. Neglecting the dependence between the variables leads to some deviations between the hydrological model results achieved using the meteorological observations and those using meteorological simulations.*

11) p.6., l.18: Does "model chains" here refer to "GCM-RCM combinations" - if yes, this term would be more precise and should be used alternatively. Or are "GCM-RCM combinations for different scenarios and realizations" the content behind the "model chains" - in any case, please be more precise and avoid the use of "model chains" as it can include quite a lot of pre- and postprocessing steps when used without further clarification.
**Reply:** *We will replace the term model chain by GCM-RCM combinations for different scenarios since a chain in our case encompasses an RCP, a GCM, and an RCM.*

12) p.6, l.19: Please rephrase to "for the locations of various meteorological stations".
**Reply:** *We will rephrase the sentence.*

13) p.6, l.23: Please add "for topographic corrections" or a similar completion to "was additionally used".
**Reply:** *We will complete the sentence as suggested.*

14) p.6, l.34: Although later specified, it is not clear whether the applied discharge series is that observed for the gauging stations or the hydrological simulations - maybe adding "(here

the simulated discharge for past and future conditions)" would make this clear as early as possible.

**Reply:** *We will specify this in the text.*

15) p.6, eq. 1: What does "i" stand for?

**Reply:** *It will be specified that i represents the imaginary unit.*

16) p.6. l.14: "Model chains" is not really precise (see comment 11), please provide additional information.

**Reply:** *We will replace the term model chain throughout the text and make it more specific.*

17) p. 7, l.24: Here the reader wonders how these unrealistic estimates are handled and how they affect the robustness of the study findings - would it be an option to already point out here that the univariate technique will not find further consideration in the study as later mentioned (see p. 11, l. 2)?

**Reply:** *This information will be added to the text at this early point of the manuscript.*

18) p. 7, l.3: Does this mean the typical flow regime assumed is allowed to be different for past and future conditions depending on the simulated discharge time series - this is an important point and needs to be included.

**Reply:** *This point will be included.*

19) p. 8, l. 6: Is the number necessarily as high as 1500 years to apply this approach or is it 1500 years because the authors carried out simulations for 1500 years (see p.7, l. 11)? Maybe adding "here" before 1500 would avoid any speculation.

**Reply:** *The approach is very flexible as to how many simulations are performed. We here run the simulation procedure for 1500 years, which will be pointed out in the text.*

20) p.8, l. 13: The authors describe that a "control regime" was generated on the basis of discharge simulations achieved using meteorological observations as input for the hydrological model and that a number of "reference regimes" have been derived from the hydrological simulations based on the different GCM-RCM combinations, resulting in a range of current regime estimates. This is not fully in line with previous statements that describe that the discharge simulations representing past discharge conditions are based on meteorological observations (see p.4, l. 9) so it would be beneficial to modify the statements at p.4, l. 9 accordingly.

**Reply:** *We will extend the statement by saying that we also simulated discharge for current conditions using meteorological input from a set of climate models.*

21) p.8, Figure 3: Please add to the caption that this figure is just a schematic illustration of the comparison approach to make clear the underlying data is not generated in this study and should not be interpreted.

**Reply:** *We will add to the caption that the figure is just an illustration.*

22) Figure 4 & 5: For me it is not quite clear whether the extreme regime estimates for this control setting are derived based on simulated discharge (see p.8, l.13) or on the discharge observations also illustrated in Figure 4 and 5. However this makes a difference, as using discharge observations for the extreme regime estimation would represent a "perfect setup" to only quantify the uncertainties in the estimation approaches, while using discharge simulations based on meteorological observations (previously defined as "control conditions") would rather illustrate the uncertainty in the whole control setting (which according to the described study design includes the uncertainties from the hydrological modelling using meteorological observations as input) as well as the uncertainties from the individual regime estimation method. Ideally, the extreme regime estimation methods would also be tested using the reference simulations (achieved using climate model data for the past as input for the hydrological model) to show the uncertainty additionally induced by applying climate simulations as hydrological model input (as done for low-flow conditions in Figure 6 using only the FDC method).

**Reply:** *Figures 4 and 5 are based on discharge derived from observed meteorological data in order to assess the usefulness of the different estimation approaches. This will be specified in the Figure and its caption. The choice of the extreme-regime estimation method was taken based on this comparison. The suitable methods (FDC and stochastic simulation) were afterwards applied to the reference simulations (hydrological model driven with simulated meteorological data). The uncertainty coming from this is shown in Figure 6. We will extend the Figure by the high-flow conditions to provide some information on the method's performances for high-flow conditions.*

23) p. 11, Figure 6: This figure is nice as it for the first and only time compares the results achieved for meteorological observations (control simulations) and the climate simulations - is there any reason this is only done for low-flow conditions? There also is some room for improvement with respect to the graphical realization: The different signatures show the "control simulations" (observed meteorological input for the hydrological model) and "reference simulations" (data from different GCM-RCM combinations as input for the hydrological model). The description "Control simulation" and "Climate simulations" is a little confusing, as for the control simulations the word simulations represents hydrological simulations, while in the case of the climate simulations the word simulations refers to the meteorological input applied. I therefore suggest denoting the signatures "control simulations" and "reference simulations" with a detailed description (as already provided) in the figure caption. Moreover, I would suggest to choose a dashed-type of signature for the upper and lower borders of the reference simulations (climate simulations) similar to the future range in Figure 7) to make them distinguishable from the control simulations.

**Reply:** *We will complement the figure with the high-flow regime estimates. The Figure legend will be adjusted as suggested by the reviewer. The borders of the polygons will be converted to dashed lines.*

24) p. 11, l.5: I would in general avoid the word "model chain" and replace it with something more precise (e.g. GCM-RCM combination). "Model chain" is a rather wide term that can

include a hydrological model, the extreme regime estimation or the downscaling and bias-correction procedure (see also comment 11).

**Reply:** *As mentioned previously, we will replace the term model chain by a more specific term throughout the document.*

25) p. 11, l. 8: Better replace "observed data" with "meteorological observations" to be more precise. I would also replace "means" with "suggests" as it is rather a hint in that direction and not a fact (also the line is in the spread range, the individual reference simulations can be quite far from the line).

**Reply:** *We will replace the words as suggested by the reviewer.*

26) p. 11, l. 9: The authors explain the overestimation in summer with an overestimation in RCM-simulated precipitation. This is plausible, but shouldn't the precipitation statistics in the RCM data match that of the observations after application of the quantile mapping approach? Maybe the authors could deepen the discussion on this issue in the updated version of the manuscript.

**Reply:** *The overall precipitation should indeed match the observations. However, precipitation and temperature have been bias corrected in a univariate manner.* Meyer et al. (2019) *recently showed that bivariate (temperature and precipitation) bias correction might be preferable in mountainous catchments where the interplay between temperature and precipitation has a significant impact on snow accumulation and therefore on the seasonality of discharge. This discussion point will be added to the manuscript.*

27) p. 12, Figure 7: I assume that the current regime is based on the hydrological simulations achieved using meteorological observations as input for the hydrological model (as described earlier in the text, see p.4, l.9). I see a certain weakness of this study in directly comparing the hydrological simulations achieved with meteorological observations (current) and climate simulations (future). While the applied bias correction fits the statistics of the climate simulations for the past to those of the observations (but also only in case of those meteorological variables that are bias corrected, see comment 10), the data sets still are not identical for the past and can be expected to induce different hydrological reactions, particularly when it comes to extreme events. I think the study would have benefitted from using a control regime that is also based on climate simulations, in this case for the past (e.g., multi-model mean of all hydrological model results achieved using different GCM-RCM combinations). Otherwise the systematic differences in the driving data make the results hard to compare. This at least needs to be discussed in the updated manuscript.

**Reply:** *Thank you for this suggestion. We agree that a comparison of the future regime estimates with an estimate derived using the control run is not ideal. We will follow your suggestion and use a multi-model mean derived from the 39 GCM-RCM combinations instead as a reference for the current climate. This multi-model mean will be used in the updated figures and to compute the differences between future and current regime estimates.*

28) p.12, Figure 7: The solid (current) and dashed lines (surrounding the shaded areas) in the case of the "mean conditions" are black in the legend but seem to be grey in the charts - this should be corrected. The caption indicates that the "normal" regimes are provided as a reference in grey, but are these the same as the "mean conditions", if yes please try to avoid using different words for the same content to make the figure easier to understand. The legend should in this case also be modified to link a grey solid line to the current state and grey dashed lines to the boundaries surrounding the grey areas as described in the caption. Finally, the areas where orange and light-blue areas overlap are extremely hard to distinguish from light-grey areas making the plots extremely hard to read. Please try to update the plots to make them easier to read - maybe there is simply too much information in them? Separating mean and extreme conditions would already make a big difference. Moreover, rather or additionally discussing the multi-model mean instead of the spread in future discharge achieved from all GCMRCM simulations could reduce the displayed information for the sake of readability and interpretability.

**Reply:** *The legend will be corrected. We will use the term mean instead of normal consistently throughout the manuscript. The readability of the figure will be improved by removing redundant axis labels to save space and by replacing the thick dashed lines around the polygons by thin dashed lines. We would prefer to not separate normal from extreme conditions since the regimes derived for mean conditions serve as a reference.*

29) p.12, l.3: I would suggest to rephrase "These chains" to "Here, the different realizations".
**Reply:** *The sentence will be rephrased.*

30) p.12, l.3: There seems to be also a distinct reduction in mean and extreme conditions for FDC in both rain dominated catchments in spring. Apart from RCP4.5 in the Jura catchment this reduction is clearly evident in both rain dominated catchments and all scenarios. Maybe this could also be an issue of discussion.
**Reply:** *This observation will be added to the text.*

31) p.12, l.8: Replace "differences in" with "differences between".
**Reply:** *The replacement will be conducted.*

32) p.12, l. 9: Why has RCP4.5 been excluded from Figure 8. If it is because you expect the changes to be in between those resulting from applying RCP2.6 and RCP8.5 please add a short sentence including this information to the text.
**Reply:** *We did not include the results for RCP 4.5 to increase the readability of the plot. The results indeed lie in between those of RCP 2.6 and RCP 8.5. A short sentence providing this information will be added to the text.*

33) p.13, l. 1: Do you mean "different" instead of "distinct" - "different" would be better from my point of view as you explain how the changes differ in the following lines.
**Reply:** *Distinct will be replaced by different.*

34) p.14, Figure 8: "Mean" is used in the plot and "normal" is used as a describing term in the caption - I would suggest using one of the terms as the counterpart to "extreme consistently in the figures and throughout the manuscript. Please rephrase "the second three rows" to "the last two rows" in the caption as there are only five rows in total.
**Reply:** *Mean will be used instead of normal consistently throughout the document. The sentence will be rephrased.*

35) p.14, l.2: I would suggest to include a brief description of the main findings from Figure 9 before moving on to Figure 10.
**Reply:** *A short sentence will be added the make the transition between Figures 9 and 10 more fluent. The detailed results are discussed after having displayed Figure 10 since it is hard to determine actual changes from Figure 9.*

36) p.14, l.3: Are the changes really "independent of the estimation technique used (FDC/stochastic)" or just in a similar order of magnitude or comparable?
**Reply:** *We agree that similar is more appropriate and will change the wording.*

37) p.15, Figure 9: As Figure 7 and Figure 9 share many characteristics, the options for improvement described for Figure 7 (comment 28) mostly also apply to Figure 9. Moreover, the y-axes in the case of the Thur catchment need adjustment towards higher maximum y-values as the maxima are cut out in the case of all scenarios.
**Reply:** *We will adjust the figure according to the changes also made for Figure 7 (see response above). The y-axes of the Thur catchment will be adjusted.*

38) p.16, Figure 10: Please rephrase "the second three rows" with "the last two rows" in the caption as there are only five rows in total.
**Reply:** *The caption will be rephrased.*

39) p.17, l.17: Better change "include future climate scenarios" to "include the assumptions underlying the applied future global climate scenarios". Maybe also change "the choice of a hydrological model" to "the uncertainties inherent in the hydrological model results".
**Reply:** *The sentences will be rephrased.*

40) p.17, l.33: Please replace "regions but the low-flow" with "regions but with the low-flow".
**Reply:** *The passage was rephrased.*

**Technical corrections (typing errors, etc.)**

1) p.2, l.6: Better rephrase to "... in the future ...".
**Reply:** *This will be rephrased.*

2) p.5, l.9: I think "glacier melt" in two separate words is the most commonly used form.
**Reply:** *The word will be changed.*

3) p.18, l.23: Please replace "which coincides" with "which coincide" and "high flow" with "high-flow".

**Reply:** *The phrasing will be changed.*

**References used in the answers to the reviewers**

Hosking, J.R.M., 1994. The four-parameter kappa distribution. IBM J. Res. Dev. 38, 251–258.

Krysanova, V., Donnelly, C., Gelfan, A., Gerten, D., Arheimer, B., Hattermann, F., Kundzewicz, Z.W., 2018. How the performance of hydrological models relates to credibility of projections under climate change. Hydrol. Sci. J. 63, 696–720. https://doi.org/10.1080/02626667.2018.1446214

Meyer, J., Kohn, I., Stahl, K., Hakala, K., Seibert, J., Cannon, A.J., 2019. Effects of univariate and multivariate bias correction on hydrological impact projections in alpine catchments. Hydrol. Earth Syst. Sci. 23, 1339–1354. https://doi.org/10.5194/hess-2018-317

Zekollari, H., Huss, M., Farinotti, D., 2019. Modelling the future evolution of glaciers in the European Alps under the EURO-CORDEX RCM ensemble. Cryosph. 13, 1125–1146. https://doi.org/10.5194/tc-13-1125-2019

---

## Author Comment (AC2) · 11 Jun 2019

**Reviewer 2**

**Overview**

This paper considers future changes in 'extreme flow regimes' in 19 hydrological regions in Switzerland as interpreted from annual flow duration curves and annual hydrographs aggregated as monthly average flows. Two novel methods are applied to generate estimates for the 100-year regimes for the current climate (using simulations driven by observed meteorological data) and a future climate (with simulations driven by bias-adjusted climate model output). The methods are illustrated using four example regions, representing both rainfall-dominated and snowmelt-dominated flow regimes, before being applied to the 19 hydrological regions. The results point towards patterns of change which are distinct for rainfall-dominated vs. snowmelt-dominated flow regimes, and which are consistent with previous work and with changes in the hydrometeorological drivers. The authors also propose that the two approaches applied give similar and consistent results.

**General comments**

Overall, I find the quality of the work presented to be quite high.

The topic addressed has important practical applications, and the introduction and application of alternative methods for interpreting changes in extreme conditions from data series of limited length (30-year) is much needed in climate change impacts research.

The manuscript is well-written, has a structure and figures that are in most cases clear and easy to follow. I am, however, in agreement with Reviewer #1 that the major weakness of the work is the comparisons made between simulations based on observations (for the current climate) with those based on climate model data (for the future climate) in order to interpret expected changes in flow regimes. One can only justify comparisons between reference and future simulations based on climate model data, due to the inevitable discrepancies between simulations based on observed data vs. climate model data. This needs to be corrected before the manuscript can be considered for publication.
**Reply:** *Thank you very much for pointing this weakness out. We will as also suggested by reviewer 1, compare the future regime estimates to a multi-model mean derived from the 39 reference simulations. The control run simulated using the observed meteorological data will not be used anymore in comparisons to the future conditions. All the affected figures and results will be updated accordingly.*

Secondly, the choice of the use of a direct stochastic simulation method rather than an 'indirect' stochastic method (which is mentioned but not discussed at all) should be presented in more detail. In particular, the use of a direct stochastic simulation of discharge from 'sampled' discharge (either simulated or observed) entails the assumption that events with long return periods come from the same population as those with shorter return periods (and can therefore be extended based on their power spectrum or using other extrapolation techniques). With 'indirect' methods, this constraint is less severe, in that

driving factors producing the flows (e.g. precipitation and initial conditions) can be modelled individually and thus will potentially produce a wider range of likely flow conditions and as of yet unobserved combinations. There are nevertheless many drawbacks with the 'indirect' methods, so the authors should use the opportunity to here to highlight why they have chosen their particular stochastic approach.

**Reply:** *The direct stochastic simulation approach used here uses a very flexible distribution (four-parameter kappa) to model the distribution of daily flows, which allows for the generation of a wide range of flow values including low and high extremes. The statistical model is therefore well able to produce extremes. Furthermore, the focus of this study was not on individual events but rather the whole regime and the reproduction of single events would therefore anyway be less relevant than for other applications especially focusing on extreme events. We think that this simple approach has the advantage of not including many uncertainty sources commonly included in hydrological modeling (e.g. choice of model, parameter equifinality). A justification for having chosen a direct instead of an indirect approach will be added to the manuscript.*

My final reservation about the manuscript as it is currently presented is that the discussion of the performance of FDC and stochastic methods is limited and rather superficial. I don't see that the two methods give uniformly similar results, particularly for high flow regimes in rainfall-dominated catchments. The magnitude of the change in maximum discharge under a future climate are both higher and lower for the stochastic method than for the FDC method and the projections give different seasons for the maximum.

**Reply:** *We agree that there are some differences between the two approaches which should be discussed in more detail. We extended the discussion section saying that the differences between the two methods mainly lie in how the seasonality is derived. In the case of the FDC approach, mean seasonality is used. In the case of the stochastic approach, a rather "random" seasonality is used since the regime is chosen according to the annual discharge sum. The direction of changes derived from the two estimates are similar except for changes of minimum discharge in the low flow regime and minimum discharge in the high flow regimes.*

In addition, results comparing simulations based on observed vs. climate data for the current climate are not shown for high flows. Given discrepancies between the two methods found in other figures, one would also like to be able to assess the correspondence between the simulations for high flows. A full development and discussion of the results, with reference to the different aspects considered here (i.e. snowmelt vs. rainfall-dominated catchments and high flow vs. low flow regimes) would significantly enhance the contribution of this work to the scientific literature. Overall, I find this to be a valuable piece of work will be worthy of publication, once the issues raised above have been addressed. Otherwise, I have only a few additional minor comments, questions and proposed corrections as given below.

**Reply:** *The validation results will also be shown for high-flow, i.e., Figure 6 in the manuscript will be completed by the high-flow regime estimates.*

**Specific comments**

P1 – Abstract: Well-presented, but at this point in the manuscript I also needed a clarification as to what is meant by 'flow regime'. Perhaps you should more clearly emphasis that in your end results you are interpreting/analysing 'flow regimes' using annual hydrographs comprised of monthly averaged flows. Particularly, for the case of high flow regimes, the time unit analysed is of interest.
**Reply:** *A specification will be added to the text.*

P2 L24-25 I don't agree with the statement that the focus on an individual characteristic "neglects the pre-conditions of low- and high-flow events", at least in terms of how most climate impact studies consider changes in this variables. For example, if one is evaluating changes in maximum daily flows based on daily simulations of 30-year periods for a present and a future climate, potentially higher flows in the period preceding the largest events are also represented in those simulations, i.e. they are not neglected. It is true that by considering an index which targets a broader time window or discharge range you are able to more directly interpret why these changes occur, e.g. because discharge is elevated for a longer time period and is thus more susceptible to extreme precipitation. In the 'standard' one characteristic approach, this would require a further analysis. A similar argument can be made for low flow indices, i.e the antecedent conditions are simulated and can be analysed if one requires additional information as to factors responsible for the estimated changes. However, to simply state that pre-conditions are neglected is misleading and incorrect.
**Reply:** *We did by no means intend to say that the use of climate simulations does not allow for the consideration of extreme events. What we intended to say was that frequency analyses often focus on one hydrological characteristics instead of considering the flow regime as a whole. Flood frequency or low flow frequency analyses typically do not explicitly consider antecedent conditions. The sentence will be rephrased to avoid confusion.*

P3 L15-20 As mentioned in the general comments, this paragraph needs to justify why direct stochastic simulation of discharge is used in this work. I would therefore suggest that the paragraph opens with a sentence describing what stochastic simulation is in general, and that this is followed by a more thorough discussion of the advantages and disadvantages of indirect vs. direct approaches, before you jump into a detailed discussion of options for direct simulation. It is important that you place the direct stochastic simulation method you have chosen within its broader context, and argue for why it is preferable and suitable for use here. In particular, there is a wide and growing literature on 'indirect' stochastic simulation which should be covered here with at least 2-3 sentences.
**Reply:** *We introduce the aim of stochastic simulation and will add a short overview on the two different main approaches: direct and indirect. Then, we will explain that we chose a direct simulation approach because it's easy to apply and works without having to progress precipitation through a hydrological model.*

P6 L17 'regional downscaling approach based on quantile mapping'. . ..more detail is needed. In particular, it is not clear in the description given in L20-L21 at what point the bias

correction with quantile mapping is applied. I assume that the same data which is used for the simulations based on observed data is used for the bias correction. . ..is this correct? In addition, what is the time period of the observed data used for bias correction?

**Reply:** *The quantile mapping was applied to simulated raw model data, which were bilinearly interpolated from the 12 or 50 km RCM resolution to the observational 2 km grid. It was performed on a grid-cell-by-grid-cell basis. The calibration was performed with the observed gridded data on the period 1981-2010, which was used to drive the model in the control run. More detail on the quantile mapping procedure will be provided in the updated version of the manuscript.*

P6 L18 . . ..'39 model chains (Table A1'. The choice of models used should be discussed, rather than leaving it to the reader to decipher a table in the appendix in order to determine how many different GCMs vs. RCMs vs. RCPs vs. grid resolutions are represented by the 39 simulations. The different models, etc. represented in the ensemble will indeed have an effect on both the mean values estimated and the spread about the mean, and the 39 model chains used here are only a subset of available EUROCORDEX simulations. In particular, it is unclear to me why both EUR-11 and EUR-44 grid resolutions have been used, if the climate model data are used for hydrological simulations at a 200-m grid resolution. I suspect that the EUR-44 simulations have been included to give a larger ensemble; however, this also means that the GCMRCM combinations that are available for both EUR-11 and EUR-44 are more heavily weighted in the ensemble. A brief (2-3 lines) discussion of the composition of the ensemble is therefore needed in the manuscript.

**Reply:** *The ensemble will be more thoroughly described and a discussion on its composition will be provided.*

P7 L13-14 Need to also generate stochastic series for the current 'conditions' using the hydrological model simulations based on the 39 model chains for the period 1981-2010.

**Reply:** *We derived stochastic simulations from the reference simulations derived from each of the 39 model chains. These were later on used to estimate the extreme regimes for the reference period and to compute a multi-model mean serving as a reference.*

P8 L2 – 'Long-term mean'. . .can be more specific, i.e. the mean regime over the period 1981-2010? I assume from your figures that this is aggregated by month?

**Reply:** *The long-term mean for the current conditions was computed over the period 1981-2010. The one for future conditions over the period 1971-2100. We specified that the regime varied for current and future conditions and that it was computed at a daily resolution but aggregated to a monthly scale to allow for a comparison between the univariate and FDC estimates.*

P8 L14-15 It is this 'reference' simulation which should actually be the 'control' simulation and be further used to evaluate future changes

**Reply:** *We agree and will use a multi-model mean reference simulation to derive the differences in regime estimates between future and current conditions.*

P8 L17 'were treated separately'. . ..not sure what is meant by this. Perhaps simply that the results were grouped by RCP?

**Reply:** *This will be rephrased as suggested.*

P10 L5-7 Why is the seasonality so variable between the FDC, Stochastic and Univariate estimates in the rainfall-dominated Thur and Jura catchments? Doesn't this undermine the credibility of these methods for considering high flows in these and other rainfall-dominated catchments?

**Reply:** *We will specify that the univariate approach produces unrealistic results in terms of seasonality, since the predictions of the monthly 100-year flows neglects the dependence between the different months. The FDC and stochastic approaches produce rather similar results excepts for the Thur catchment. While the FDC approach works with a mean seasonality, the stochastic approach uses the seasonality of one specific realization (annual hydrograph), whose discharge sum corresponds to the 100-year discharge sum.*

P10 L10-12 Given my comment above, I am also curious as to what Fig. A1 looks like for high-flow regime estimates? In particular, the stochastic results for Jura in Fig. 5 suggest that the method doesn't perform better than the univariate approach in that case.

**Reply:** *The stochastic approach is not supposed to exactly reproduce the mean seasonality reflected by the FDC approach. While the seasonality generated by the univariate approach is something completely artificial, the seasonality produced by the stochastic approach could have been observed. Figure A1 was also produced for the high-flow estimates and is provided here for completeness.*

[Figure]

**Figure 1: Comparison of the 100-year high-flow estimates, univariate, FDC, and stochastic with stochastically generated hydrographs (orange lines). The legend is equivalent to the one provided for Figure A1 in the manuscript.**

P11 Sec 3 Would also like to see a comparison of the climate simulations and the control simulation for high flows, similar to that shown for Fig 6 for low flow regimes.
**Reply:** *Figure 6 will be extended by the high-flow regimes.*

P12 L10-11: Changes in mean flow of up to 50%, etc. These estimates are not reliable because you are comparing the results of simulations based on climate model data (future) with those based on observations (current), and thus are also including some the error illustrated in Figure 6 in your estimates.
**Reply:** *We newly computed the differences between current and future regime estimates by using the multi-model mean derived from the 39 reference simulations as a reference estimate. The text was adjusted accordingly.*

P17 Section 4.1: In addition to discussing the overall merits of the two methods, it would also be useful to see a discussion of their relative performance for high vs. low flow regimes and for rainfall-dominated vs. snowmelt-dominated catchments. I also find that the second paragraph in this section is too general and should focus on discussing and expanding the results you have presented in more depth. It would also be useful, for example, if you could highlight aspects of the methods you have used which are better for quantifying changes in flow regimes useful for management purposes, relative to those commonly used for climate impact studies.

**Reply:** *The differences between the estimates derived using the FDC approach and those derived using the stochastic approach will be discussed in more detail. However, it is difficult to say which one is the "better" method since the "true" 100-regime estimate is not known.*

P18 L15-16: 'Both were found to provide realistic, mutually-agreeing results'. I think that this is bit overstated. In particular, the results for the high flow regimes for the 19 regions (Fig. 10) show a similar direction of change (in most cases), but the magnitude is in some cases much higher with the Stochastic method, and this would have significant implications for their application in practice.
**Reply:** *The statement will be weakened by replacing mutually-agreeing with similar.*

**Technical comments**

P1 – Keypoints: (L22) – 'are changing' should be 'will change', i.e. you have not examined patterns of change under current conditions, which is what the English used here implies
**Reply:** *The keypoint will be rephrased.*

P2 L17-18: Considering rephrasing, for example to: For planning purposes and river basin management, however, estimates not only for normal conditions but also for extreme conditions are needed.
**Reply:** *The sentence will be rephrased.*

P4 L3. . .should add 'under the current climate' after 'regime.'
**Reply:** *We will specify this.*

P12 L4 Replace 'expressed' with 'pronounced'
**Reply:** *Expressed will be replaced by pronounced.*

P14 Fig 8; P16 Fig10 caption: In line 3 should be 'The top three rows show relative changes, and the bottom two rows show changes in months'
**Reply:** *The captions will be adjusted.*

P18 L8-9: Replace 'We here showed' with 'We have shown here'
**Reply:** *The sentence will be rephrased.*

**References used in the answers to the reviewers**

Hosking, J.R.M., 1994. The four-parameter kappa distribution. IBM J. Res. Dev. 38, 251–258.

Krysanova, V., Donnelly, C., Gelfan, A., Gerten, D., Arheimer, B., Hattermann, F., Kundzewicz, Z.W., 2018. How the performance of hydrological models relates to credibility of projections under climate change. Hydrol. Sci. J. 63, 696–720. https://doi.org/10.1080/02626667.2018.1446214

Meyer, J., Kohn, I., Stahl, K., Hakala, K., Seibert, J., Cannon, A.J., 2019. Effects of univariate and multivariate bias correction on hydrological impact projections in alpine catchments. Hydrol. Earth Syst. Sci. 23, 1339–1354. https://doi.org/10.5194/hess-2018-317

Zekollari, H., Huss, M., Farinotti, D., 2019. Modelling the future evolution of glaciers in the European Alps under the EURO-CORDEX RCM ensemble. Cryosph. 13, 1125–1146. https://doi.org/10.5194/tc-13-1125-2019

---

## Editor Comment (EC1) · Axel Bronstert (Editor) · 9 Jul 2019

Dear authors, thank you very much for revising and improving your manuscript. I think you did a good job. However, it needs to go through a 2nd round of reviews. I agree with both reviewers, that the model-based assessments of hydrological "impacts" of climate change needs to show the performance or the climate-water-model system for the current conditions before applying for future (scenario) conditions. As far as I understood, you now changed the current situation assessment by using an ensemble of the RCMs for reference conditions instead of driving with observed meteorological conditions. I recommend to show BOTH such results, and to discuss how large the differences are (in particular in respect to the "signal" of climate change scenarios), and how relevant you consider these differences and how you coped with those. Kind

regards Axel Bronstert (editor of this manuscript)

---

## Author Comment (AC3) · 10 Jul 2019

Dear Prof. Bronstert,

Thank you for your comment. We completely agree that the hydrological model used for future simulations needs to be validated first for current conditions. We did this by first driving the model using meteorological observations. In a second step, we drove the model with meteorological data simulated using a GCM-RCM model chain. The hydrological regimes derived in the first step (observed meteorology) were compared to the hydrological regimes derived in the second step (simulated meteorology). Figure 6 in the revised manuscript shows that the model and the simulated meteorology are well able to reproduce the regimes derived from the simulations generated using the

observed meteorology except for one region (i.e. Engadin) where the low-flow regime is overestimated when using the simulated meteorology. Given these validation results, we are convinced that the model is suitable for answering our research questions.

Best regards, Manuela Brunner

---

## Author Response (AR1)

**Dear Prof. Bronstert,**

We thank you and the reviewers for acknowledging the value of our work and appreciate the constructive comments and suggestions for improvement. The revised version of our manuscript addresses the main point of criticism regarding the use of a control vs. the use of a reference regime. We agree with the reviewers that the use of reference regime estimates derived from the 39 GCM-RCM combinations is indeed more suitable than the use of a control regime derived based on observed meteorology. The methodology and results were adjusted accordingly.

Below, we address the points risen by the two anonymous reviewers and state how we would like to address them in a revised version of the manuscript. Our replies to the reviewers' comments are written in blue and italic to distinct them from the reviewers' comments.

On the behalf of all co-authors,

Yours sincerely,

Manuela Brunner

**Reviewer 1**

**General comments (overall quality of the discussion paper)**

The authors have presented a nicely prepared study that applies hydrological simulations driven by different meteorological forcings for past and future climate conditions to estimate changes in high- and low-flow conditions for different rain- and melt-dominated catchments in Switzerland. Thereby, different approaches for estimating high- and lowflow regimes are applied and compared, discussing the individual strengths and weaknesses of each approach. A total number of 39 realizations of three different IPCC representative concentration pathways (8 x RCP2.6, 13 x RCP4.5 and 18 x RCP8.5) is compared to reflect the bandwidth of potential change in future discharge conditions in the catchments considered. As the knowledge on potential changes in highland low-flow conditions is of high societal, economic and ecological importance, the present study is of high scientific relevance with the thematic focus fitting nicely in the thematic scope of HESS. The formal requirements for publishing are almost fully met as reflected in the low number of technical corrections suggested in the following.

My major concerns relate to the comparisons of current and future discharge conditions. While the hydrological simulations representing the data basis for the estimation of current flow conditions are driven by meteorological observations, the simulations for future discharge conditions are based on climate conditions simulated by different GCM-RCM combinations. Although the downscaling approach of quantile mapping has been applied to

statistically correct biases in the climate simulations for past and future conditions, the meteorological conditions in the observational data set applied for the hydrological simulations for the past are still not identical to the RCM simulated climate simulations for the past. This induces biases when it comes to the comparison of past and future discharge conditions calculated on the basis of meteorological data from the two different origins. Differences in the meteorological data sets can be expected to occur at shorter time scales (as highly relevant for extreme weather and hydrological conditions) as well as with respect to the interdependency of various meteorological variables, e.g. temperature, humidity and precipitation (such inter-variable consistency is not conserved applying quantile mapping). The existing differences in the meteorological observations and climate simulations with their respective hydrological effects for the past become clearly evident looking at the differences between the hydrological simulations achieved using meteorological observations and simulations in the case of both mean and extreme conditions in Figure 6. Moreover, calibration of the hydrological model seems to be carried out using meteorological observations for the past. While less important compared to the issue described before, calibration can indirectly compensate deficiencies in the meteorological input applied during calibration. Applying the same calibrated parameter set using different meteorological input (with very likely different deficiencies, e.g. those related to the quantification of precipitation) might lead to inconsistencies in the model results. A way to avoid the inconsistencies arising from different meteorological input for the past and future would be to correct the RCM simulations for the past and future statistically (as done in the present study) and later only compare the differences between the hydrological simulations for the past and the future using past and future climate simulations as input for the hydrological simulations. Figure 6 follows this direction but all other results seem to compare current and future conditions generated on the basis of different meteorological input (station observations vs RCM simulations). Using the just outlined approach the hydrological simulations would be better comparable as (hopefully) the same systematic biases are found in the simulations for the past and future. As a large number of RCM simulations are providing meteorological conditions for the past in this study, this would require defining some sort of "one hydrological reference" (e.g., the multimodel mean) the different scenario simulations can then be compared to. Apart from this main point of criticism, some further points for improvement remain, which are described in the "Specific comments" and mostly represent suggestions for clarification as well as options to make the contents of the study easier transportable to the reader.

A final issue to be mentioned is, that at some point a closer linkage of the hydrological results to the climate change signal in the applied climate scenarios for Switzerland (e.g., by showing and discussing the temperature and precipitation change) would be beneficial to interpret the presented hydrological changes. However, considering the large number of scenarios and realizations this is probably beyond the scope of the article and should hence not be an issue that needs to be addressed in the revision.

As a final recommendation, I suggest to accept the article for publication in HESS given all issues have been addressed properly (major revisions)

**Reply:** *Thank you for your suggestion of replacing the regimes derived from a control run (observed meteorology) by regimes derived from a reference run (simulated meteorology). We followed your suggestion and replaced the regimes derived from the control run by a multi-model mean computed from the 39 reference runs. All the figures affected were updated accordingly.*
**Modification: p:9, l:7-9, Figures 7-10**

**Specific comments (individual scientific questions/issues)**

1) p.3, l.24: The authors describe existing approaches for the generation of discharge time series and then name the approach chosen in their study. Although more detailed information is provided in the "Methods section", it might be valuable for the reader to receive some brief description, in how far the applied approach differs from or matches the ones described shortly before. Maybe the authors could add some information on this here?
**Reply:** *We added a brief description of the differences of the simulation approach employed here as compared to existing approaches. As opposed to classical phase randomization approaches, this approach does not rely on the empirical distribution but uses the flexible, four-parameter kappa distribution (Hosking, 1994), which allows for the generation of a wide range of realizations of high and low discharge values.*
**Modification: p:3, l:28-30**

2) p.4, Figure 1: This figure is considered very important for the reader to get an overview of the study regions. However, apart from the outlines of the study regions, the map seems rather general and would benefit from some additional detail. Maybe it would be possible to support orientation for non-EU readers e.g., by modifying Figure 1, including a larger scale overview map linked to the original map as well as by adding some more details e.g., the names of larger rivers, lakes or cities?
**Reply:** *We added an inlet map of Europe, showing the location of Switzerland within Europe. Since the map is already quite busy restricted the labels to the ones already present.*
**Modification: Figure 1**

3) p.4, l.7: The authors include the information "second step" in brackets - this approach helps to structure the workflow and might be a good extension also for the first and following steps, which are currently referred to as part of the describing text. Having all the steps in brackets would make it easier to navigate through the workflow for the reader.
**Reply:** *We added the number of the step in brackets.*
**Modification: p:4, l:11&14**

4) p.4, l.9: The discharge series used for the estimation of extreme regimes are based on hydrological simulations using observed meteorological conditions for past discharge conditions and simulated meteorological conditions (climate model output) for future

discharge conditions. Given the biases in currently available climate simulations the reader here wonders if some correction has been applied before the application of the climate simulations for past and future in this study. Although this is later explained adding "bias-corrected" and "downscaled" might satisfy the readers curiosity at this early point.

**Reply:** *The information was added to the text.*

**Modification: p:4, l:17**

5) p. 5, Figure 2: Number 1 in the workflow (comparison procedures) leads to some confusion from my point of view. I understand from the workflow description starting on p.4, l.5 that the different methods for estimating extreme flow regimes were tested, however this comparison seems to require the hydrological simulations of the left side of Figure 2 as input already, which somehow conflicts with the rank in the overall workflow.

Some further confusion in Figure 2 from my point of view arises in the context of the "Data" column. The caption says "A" introduces the simulated data used, however the two plots rather seem to show the meteorological input applied (climate simulations and observations) for the discharge simulations. On the other hand, the data flow from "A" to "C" (the "Estimation" box), as well as that from "A" to "B" requires the two data inputs from "A" to be the discharge simulations achieved on the basis of the two different meteorological data sources (climate simulations and observations). Maybe the authors could clarify these issues in an updated version of Figure 2. It thereby would be an option to use different colors for meteorological input and hydrological model results - while white and orange are used in the figure, the caption does not clarify whether the colors are used in this sense. Maybe it would also be beneficial to use different signatures/colors for the current and future estimates in box "C" and more clearly link them to the discharge simulations based on meteorological observations and climate simulations following the argumentation on p.4, l.9? Maybe also the PREVAH model, which is later mentioned to be used for the hydrological simulations, should be already integrated in the figure to complete the workflow?

**Reply:** *Thank you for your suggestions. Figure 2 is divided into two parts. The "Data" part and the "Estimates" part. We consider the model simulations to be part of the data generation since the main focus and innovation of our manuscript lies on the "Estimation" part. We did therefore not include the hydrological model into the figure. When we talk about data in this figure, we refer to discharge data, be it observed or simulated. This was specified to avoid confusion.*

**Modification: Figure 2**

6) p.5, l.12: The authors describe that validation was performed for the period 1983-2005, presumably using meteorological observations as input for the PREVAH model. As climate simulations are used for the model runs for future conditions, I wonder if any validation using climate simulations for the past has been carried out beside the results shown in Figure 6 (e.g., by using hindcast simulations that reproduce weather conditions and allow a comparison of simulated and observed discharge at daily basis)?

**Reply:** *The hydrological model was indeed validated using meteorological observations as input for the PREVAH model (see p. 6, l:7-9). As suggested in one of the next comments, we compared the FDCs for the reference period 1981-2010 derived based on observed meteorology to FDCs derived for the same period using the simulated meteorological data generated by the 39 GCM-RCM combinations. This analysis showed that the FDCs are reproduced well in most catchments, except for the catchment Engadin. As shown in Figure 6, the model was also validated with respect to the reproduction of past low- and high-flow regimes as these were the focus of this study. We extended Figure 6 by including the high-flow results.*

**Modification: Figure 6**

7) p.5, l.12: The authors explain that the applied hydrological model has been calibrated for the period 1993-1997. Assuming that the model was driven by meteorological observations while calibrating, the question of systematic biases due to calibration under observed meteorological conditions as well as the application of different meteorological input for the past and future arises. From my perspective both the application of different meteorological input for the past and future (meteorological stations vs climate model simulations) as well as the fact that calibration has been carried out for past observed climate conditions, whereas the simulations are presumably carried out with the same parameter set for future simulated climate conditions, need some discussion.

**Reply:** *It has been confirmed by Krysanova et al. (2018) who did a review study on the performance of hydrological models under climate change that a good performance of hydrological models in the historical period increases confidence in projected impacts under climate change. We found that the hydrological model well reproduces the hydrological regimes analyzed in this study in the reference period and therefore assume that it will also reliably simulate hydrological regimes under future conditions. A short discussion about this issue was added to the manuscript.*

**Modification: p:6, l:9-13**

It might be the case that calibration compensates for some of the biases in the observed meteorological input data (e.g. an undercatch in precipitation), which are not present or at least different in the case of meteorological simulations. Particularly in snow dominated regions differences in temperature and precipitation between the two data sources can lead to different water storage in the snow pack affecting the simulated discharge conditions. A comparison of the statistical discharge characteristics achieved for the past using meteorological observations (for which the model was calibrated) to those achieved for the past using climate simulations (which were not used in the calibration) over a climatological period of time could show the differences in the discharge characteristics (e.g., using flow duration curves), which might somehow also affect the comparability of past and future discharge simulations in the present study. Maybe the authors could add some discussion on these issues to the manuscript.

**Reply:** *As shown in Figure 6 in the manuscript, the water balance of most catchments is well represented by using the simulated meteorological data instead of the observed ones. We*

*followed your suggestion and did the validation also on the FDCs directly (Figure 1). The results show that the FDCs are mostly well reproduced by using the simulated meteorological data to simulate discharge. An exception is, as in Figure 6, the Engadin, where high flows are slightly overrepresented. We now discuss in the manuscript that this overestimation might be related to the univariate bias correction applied which might not perfectly reflect the interplay between temperature and precipitation and therefore the timing of snowmelt processes* (Meyer et al., 2019).

**Modification: p:12, l:8-10**

[Figure]

**Figure 1: FDCs derived from the control run over the period 1981-2010 (black line) and the 39 reference simulations for the same period (grey lines) for the four catchments Thur, Jura, Valais, and Engadin.**

8) p.6, l.2: Two glacier models are applied depending on the length of the glacier considered. It would be interesting why the extended GloGEMflow model is not applied to smaller glaciers (I understand that glacier flow is less important in this case but are there any reasons applying only the GloGEMflow model for all lengths is inadequate?) and what the differences between the results of both models would be - the latter might give an impression of the impact on the overall results induced by using the two different glacier models.

**Reply:** *For small glaciers a simplified glacier model is indeed used (vs. ice-dynamic model for glaciers >1 km). As shown in the original publication describing the glacier model* (Zekollari et al., 2019)*, for small glaciers the difference between the modeled glacier evolution with the simplified model and the ice-dynamic model is very small though - the difference is found to increase with increasing glacier elevation range (see Figure 12 in* Zekollari et al., 2019).

9) p.6, l.11: Does radiation refer to shortwave or longwave radiation or both? Please clarify.
**Reply:** *We clarified that radiation refers to shortwave radiation.*
**Modification: p:6, l:22**

10) p.6, l.11 and l.17: Has the approach of Quantile Mapping been applied for all meteorological variables listed in l. 11? Particularly short- and longwave radiation recordings are often not available far back in the past, reducing the possibility to statistically correct the simulations for these variables. Not correcting all variables used as input for the hydrological model would explain differences between the hydrological model results achieved using meteorological observations and those based on meteorological simulations.
**Reply:** *The quantile mapping approach was applied for all the variables. This was specified in the text. Neglecting the dependence between the variables leads to some deviations between the hydrological model results achieved using the meteorological observations and those using meteorological simulations.*
**Modification: p:6, l:32**

11) p.6., l.18: Does "model chains" here refer to "GCM-RCM combinations" - if yes, this term would be more precise and should be used alternatively. Or are "GCM-RCM combinations for different scenarios and realizations" the content behind the "model chains" - in any case, please be more precise and avoid the use of "model chains" as it can include quite a lot of pre- and postprocessing steps when used without further clarification.
**Reply:** *We replaced the term model chain by GCM-RCM combinations for different scenarios since a chain in our case encompasses an RCP, a GCM, and an RCM.*
**Modification: the term model chain was replaced throughout the text**

12) p.6, l.19: Please rephrase to "for the locations of various meteorological stations".
**Reply:** *We rephrased the sentence.*
**Modification: p:6, l:34**

13) p.6, l.23: Please add "for topographic corrections" or a similar completion to "was additionally used".
**Reply:** *We completed the sentence as suggested.*
**Modification: p:7, l:5**

14) p.6, l.34: Although later specified, it is not clear whether the applied discharge series is that observed for the gauging stations or the hydrological simulations - maybe adding "(here the simulated discharge for past and future conditions)" would make this clear as early as possible.
**Reply:** *We specified this in the text.*
**Modification: p:7, l:16**

15) p.6, eq. 1: What does "i" stand for?
**Reply:** *We specified that i represents the imaginary unit.*
**Modification: p:7, l:20**

16) p.6. l.14: "Model chains" is not really precise (see comment 11), please provide additional information.
**Reply:** *We replaced the term model chain throughout the text and made it more specific.*

17) p. 7, l.24: Here the reader wonders how these unrealistic estimates are handled and how they affect the robustness of the study findings - would it be an option to already point out here that the univariate technique will not find further consideration in the study as later mentioned (see p. 11, l. 2)?
**Reply:** *This information was added to the text at this early point of the manuscript.*
**Modification: p:8, l:10-11**

18) p. 7, l.3: Does this mean the typical flow regime assumed is allowed to be different for past and future conditions depending on the simulated discharge time series - this is an important point and needs to be included.
**Reply:** *This point was included.*
**Modification: p:8, l:21-22**

19) p. 8, l. 6: Is the number necessarily as high as 1500 years to apply this approach or is it 1500 years because the authors carried out simulations for 1500 years (see p.7, l. 11)? Maybe adding "here" before 1500 would avoid any speculation.
**Reply:** *The approach is very flexible as to how many simulations are performed. We here run the simulation procedure for 1500 years, which was pointed out in the text.*
**Modification: p:8, l:25**

20) p.8, l. 13: The authors describe that a "control regime" was generated on the basis of discharge simulations achieved using meteorological observations as input for the hydrological model and that a number of "reference regimes" have been derived from the hydrological simulations based on the different GCM-RCM combinations, resulting in a range of current regime estimates. This is not fully in line with previous statements that describe that the discharge simulations representing past discharge conditions are based on meteorological observations (see p.4, l. 9) so it would be beneficial to modify the statements at p.4, l. 9 accordingly.
**Reply:** *We extended the statement by saying that we also simulated discharge for current conditions using meteorological input from a set of climate models.*
**Modification: p:9, l:5-6**

21) p.8, Figure 3: Please add to the caption that this figure is just a schematic illustration of the comparison approach to make clear the underlying data is not generated in this study and should not be interpreted.
**Reply:** *We added to the caption that the figure is just an illustration.*
**Modification: Caption Figure 3**

22) Figure 4 & 5: For me it is not quite clear whether the extreme regime estimates for this control setting are derived based on simulated discharge (see p.8, l.13) or on the discharge

observations also illustrated in Figure 4 and 5. However this makes a difference, as using discharge observations for the extreme regime estimation would represent a "perfect setup" to only quantify the uncertainties in the estimation approaches, while using discharge simulations based on meteorological observations (previously defined as "control conditions") would rather illustrate the uncertainty in the whole control setting (which according to the described study design includes the uncertainties from the hydrological modelling using meteorological observations as input) as well as the uncertainties from the individual regime estimation method. Ideally, the extreme regime estimation methods would also be tested using the reference simulations (achieved using climate model data for the past as input for the hydrological model) to show the uncertainty additionally induced by applying climate simulations as hydrological model input (as done for low-flow conditions in Figure 6 using only the FDC method).

**Reply:** *Figures 4 and 5 are based on discharge derived from observed meteorological data in order to assess the usefulness of the different estimation approaches. This will be specified in the Figures and their captions. The choice of the extreme-regime estimation method was taken based on this comparison. The suitable methods (FDC and stochastic simulation) were afterwards applied to the reference simulations (hydrological model driven with simulated meteorological data). The uncertainty coming from this is shown in Figure 6. We extended the Figure by the high-flow conditions to provide some information on the method's performances for high-flow conditions.*

**Modification: Figures 4-6**

23) p. 11, Figure 6: This figure is nice as it for the first and only time compares the results achieved for meteorological observations (control simulations) and the climate simulations - is there any reason this is only done for low-flow conditions? There also is some room for improvement with respect to the graphical realization: The different signatures show the "control simulations" (observed meteorological input for the hydrological model) and "reference simulations" (data from different GCM-RCM combinations as input for the hydrological model). The description "Control simulation" and "Climate simulations" is a little confusing, as for the control simulations the word simulations represents hydrological simulations, while in the case of the climate simulations the word simulations refers to the meteorological input applied. I therefore suggest denoting the signatures "control simulations" and "reference simulations" with a detailed description (as already provided) in the figure caption. Moreover, I would suggest to choose a dashed-type of signature for the upper and lower borders of the reference simulations (climate simulations) similar to the future range in Figure 7) to make them distinguishable from the control simulations.

**Reply:** *We complemented the figure with the high-flow regime estimates. The Figure legend was adjusted as suggested by the reviewer. The borders of the polygons were converted to dashed lines.*

**Modification: Figure 6**

24) p. 11, l.5: I would in general avoid the word "model chain" and replace it with something more precise (e.g. GCM-RCM combination). "Model chain" is a rather wide term that can

include a hydrological model, the extreme regime estimation or the downscaling and bias-correction procedure (see also comment 11).

**Reply:** *As mentioned previously, we replaced the term model chain by a more specific term throughout the document.*

25) p. 11, l. 8: Better replace "observed data" with "meteorological observations" to be more precise. I would also replace "means" with "suggests" as it is rather a hint in that direction and not a fact (also the line is in the spread range, the individual reference simulations can be quite far from the line).

**Reply:** *We replaced the words as suggested by the reviewer.*

**Modification: p:12, l:6**

26) p. 11, l. 9: The authors explain the overestimation in summer with an overestimation in RCM-simulated precipitation. This is plausible, but shouldn't the precipitation statistics in the RCM data match that of the observations after application of the quantile mapping approach? Maybe the authors could deepen the discussion on this issue in the updated version of the manuscript.

**Reply:** *The overall precipitation should indeed match the observations. However, precipitation and temperature have been bias corrected in a univariate manner.* Meyer et al. (2019) *recently showed that bivariate (temperature and precipitation) bias correction might be preferable in mountainous catchments where the interplay between temperature and precipitation has a significant impact on snow accumulation and therefore on the seasonality of discharge. This discussion point was added to the manuscript.*

**Modification: p:12, l:8-9**

27) p. 12, Figure 7: I assume that the current regime is based on the hydrological simulations achieved using meteorological observations as input for the hydrological model (as described earlier in the text, see p.4, l.9). I see a certain weakness of this study in directly comparing the hydrological simulations achieved with meteorological observations (current) and climate simulations (future). While the applied bias correction fits the statistics of the climate simulations for the past to those of the observations (but also only in case of those meteorological variables that are bias corrected, see comment 10), the data sets still are not identical for the past and can be expected to induce different hydrological reactions, particularly when it comes to extreme events. I think the study would have benefitted from using a control regime that is also based on climate simulations, in this case for the past (e.g., multi-model mean of all hydrological model results achieved using different GCM-RCM combinations). Otherwise the systematic differences in the driving data make the results hard to compare. This at least needs to be discussed in the updated manuscript.

**Reply:** *Thank you for this suggestion. We agree that a comparison of the future regime estimates with an estimate derived using the control run is not ideal. We followed your suggestion and used a multi-model mean derived from the 39 GCM-RCM combinations instead as a reference for the current climate. This multi-model mean was used in the updated figures and to compute the differences between future and current regime*

*estimates.*

**Modification: Figures 7-10**

28) p.12, Figure 7: The solid (current) and dashed lines (surrounding the shaded areas) in the case of the "mean conditions" are black in the legend but seem to be grey in the charts - this should be corrected. The caption indicates that the "normal" regimes are provided as a reference in grey, but are these the same as the "mean conditions", if yes please try to avoid using different words for the same content to make the figure easier to understand. The legend should in this case also be modified to link a grey solid line to the current state and grey dashed lines to the boundaries surrounding the grey areas as described in the caption. Finally, the areas where orange and light-blue areas overlap are extremely hard to distinguish from light-grey areas making the plots extremely hard to read. Please try to update the plots to make them easier to read - maybe there is simply too much information in them? Separating mean and extreme conditions would already make a big difference. Moreover, rather or additionally discussing the multi-model mean instead of the spread in future discharge achieved from all GCMRCM simulations could reduce the displayed information for the sake of readability and interpretability.

**Reply:** *The legend was corrected. We used the term mean instead of normal consistently throughout the manuscript. The readability of the figure was improved by removing redundant axis labels to save space and by replacing the thick dashed lines around the polygons by thin dashed lines. We did not separate normal from extreme conditions since the regimes derived for mean conditions serve as a reference.*

**Modification: Figure 7**

29) p.12, l.3: I would suggest to rephrase "These chains" to "Here, the different realizations".

**Reply:** *The sentence was rephrased.*

**Modification: p:13, l:6**

30) p.12, l.3: There seems to be also a distinct reduction in mean and extreme conditions for FDC in both rain dominated catchments in spring. Apart from RCP4.5 in the Jura catchment this reduction is clearly evident in both rain dominated catchments and all scenarios. Maybe this could also be an issue of discussion.

**Reply:** *This observation was added to the text.*

**Modification: p:13, l:6-8**

31) p.12, l.8: Replace "differences in" with "differences between".

**Reply:** *The replacement was conducted.*

**Modification: p:14, l:1**

32) p.12, l. 9: Why has RCP4.5 been excluded from Figure 8. If it is because you expect the changes to be in between those resulting from applying RCP2.6 and RCP8.5 please add a short sentence including this information to the text.

**Reply:** *We did not include the results for RCP 4.5 to increase the readability of the plot. The results indeed lie in between those of RCP 2.6 and RCP 8.5. A short sentence providing this*

*information was added to the text.*
**Modification: p:14, l:2-3**

33) p.13, l. 1: Do you mean "different" instead of "distinct" - "different" would be better from my point of view as you explain how the changes differ in the following lines.
**Reply:** *Distinct was replaced by different.*
**Modification: p:14, l:6**

34) p.14, Figure 8: "Mean" is used in the plot and "normal" is used as a describing term in the caption - I would suggest using one of the terms as the counterpart to "extreme consistently in the figures and throughout the manuscript. Please rephrase "the second three rows" to "the last two rows" in the caption as there are only five rows in total.
**Reply:** *Mean was used instead of normal consistently throughout the document. The caption was rephrased.*
**Modification: Caption Figure 8**

35) p.14, l.2: I would suggest to include a brief description of the main findings from Figure 9 before moving on to Figure 10.
**Reply:** *A short sentence was added the make the transition between Figures 9 and 10 more fluent. The detailed results are discussed after having displayed Figure 10 since it is hard to determine actual changes from Figure 9.*
**Modification: p:15, l:1**

36) p.14, l.3: Are the changes really "independent of the estimation technique used (FDC/stochastic)" or just in a similar order of magnitude or comparable?
**Reply:** *We agree that similar is more appropriate and will change the wording.*
**Modification: p:15, l:4**

37) p.15, Figure 9: As Figure 7 and Figure 9 share many characteristics, the options for improvement described for Figure 7 (comment 28) mostly also apply to Figure 9. Moreover, the y-axes in the case of the Thur catchment need adjustment towards higher maximum y-values as the maxima are cut out in the case of all scenarios.
**Reply:** *We adjusted the figure according to the changes also made for Figure 7 (see response above). The y-axes of the Thur catchment were adjusted.*
**Modification: Figure 9**

38) p.16, Figure 10: Please rephrase "the second three rows" with "the last two rows" in the caption as there are only five rows in total.
**Reply:** *The caption was rephrased.*
**Modification: Caption Figure 10**

39) p.17, l.17: Better change "include future climate scenarios" to "include the assumptions underlying the applied future global climate scenarios". Maybe also change "the choice of a hydrological model" to "the uncertainties inherent in the hydrological model results".

**Reply:** *The sentences were rephrased.*
**Modification: p:18, l:23-24**

40) p.17, l.33: Please replace "regions but the low-flow" with "regions but with the low-flow".
**Reply:** *The passage was rephrased.*
**Modification: p:19, l:7**

**Technical corrections (typing errors, etc.)**

1) p.2, l.6: Better rephrase to "... in the future ...".
**Reply:** *This was rephrased.*
**Modification: p:2, l:6**

2) p.5, l.9: I think "glacier melt" in two separate words is the most commonly used form.
**Reply:** *The word was changed.*
**Modification: p:6, l:3**

3) p.18, l.23: Please replace "which coincides" with "which coincide" and "high flow" with "high-flow".
**Reply:** *The phrasing was changed.*
**Modification: p:19, l:6**

**Reviewer 2**

**Overview**

This paper considers future changes in 'extreme flow regimes' in 19 hydrological regions in Switzerland as interpreted from annual flow duration curves and annual hydrographs aggregated as monthly average flows. Two novel methods are applied to generate estimates for the 100-year regimes for the current climate (using simulations driven by observed meteorological data) and a future climate (with simulations driven by bias-adjusted climate model output). The methods are illustrated using four example regions, representing both rainfall-dominated and snowmelt-dominated flow regimes, before being applied to the 19 hydrological regions. The results point towards patterns of change which are distinct for rainfall-dominated vs. snowmelt-dominated flow regimes, and which are consistent with previous work and with changes in the hydrometeorological drivers. The authors also propose that the two approaches applied give similar and consistent results.

**General comments**

Overall, I find the quality of the work presented to be quite high.

The topic addressed has important practical applications, and the introduction and application of alternative methods for interpreting changes in extreme conditions from data series of limited length (30-year) is much needed in climate change impacts research.

The manuscript is well-written, has a structure and figures that are in most cases clear and easy to follow. I am, however, in agreement with Reviewer #1 that the major weakness of the work is the comparisons made between simulations based on observations (for the current climate) with those based on climate model data (for the future climate) in order to interpret expected changes in flow regimes. One can only justify comparisons between reference and future simulations based on climate model data, due to the inevitable discrepancies between simulations based on observed data vs. climate model data. This needs to be corrected before the manuscript can be considered for publication.
**Reply:** *Thank you very much for pointing this weakness out. We compared, as also suggested by reviewer 1, the future regime estimates to a multi-model mean derived from the 39 reference simulations. The control run simulated using the observed meteorological data were not be used anymore in comparisons to the future conditions. All the affected figures and results were updated accordingly.*
**Modification: p:9, l:7-9, Figures 7-10**

Secondly, the choice of the use of a direct stochastic simulation method rather than an 'indirect' stochastic method (which is mentioned but not discussed at all) should be presented in more detail. In particular, the use of a direct stochastic simulation of discharge from 'sampled' discharge (either simulated or observed) entails the assumption that events with long return periods come from the same population as those with shorter return periods (and can therefore be extended based on their power spectrum or using other extrapolation techniques). With 'indirect' methods, this constraint is less severe, in that driving factors producing the flows (e.g. precipitation and initial conditions) can be modelled individually and thus will potentially produce a wider range of likely flow conditions and as of yet unobserved combinations. There are nevertheless many drawbacks with the 'indirect' methods, so the authors should use the opportunity to here to highlight why they have chosen their particular stochastic approach.
**Reply:** *The direct stochastic simulation approach used here uses a very flexible distribution (four-parameter kappa) to model the distribution of daily flows, which allows for the generation of a wide range of flow values including low and high extremes. The statistical model is therefore well able to produce extremes. Furthermore, the focus of this study was not on individual events but rather the whole regime and the reproduction of single events would therefore anyway be less relevant than for other applications especially focusing on extreme events. We think that this simple approach has the advantage of not including many uncertainty sources commonly included in hydrological modeling (e.g. choice of model, parameter equifinality). A justification for having chosen a direct instead of an indirect approach was added to the manuscript.*
**Modification: p:3, l:15-19**

My final reservation about the manuscript as it is currently presented is that the discussion of the performance of FDC and stochastic methods is limited and rather superficial. I don't see that the two methods give uniformly similar results, particularly for high flow regimes in rainfall-dominated catchments. The magnitude of the change in maximum discharge under a future climate are both higher and lower for the stochastic method than for the FDC method and the projections give different seasons for the maximum.

**Reply:** *We agree that there are some differences between the two approaches which are now discussed in more detail. We extended the discussion section saying that the differences between the two methods mainly lie in how the seasonality is derived. In the case of the FDC approach, mean seasonality is used. In the case of the stochastic approach, a rather "random" seasonality is used since the regime is chosen according to the annual discharge sum. The direction of changes derived from the two estimates are similar except for changes of minimum discharge in the low flow regime and minimum discharge in the high flow regimes.*

**Modification: p:18, l:5-10**

In addition, results comparing simulations based on observed vs. climate data for the current climate are not shown for high flows. Given discrepancies between the two methods found in other figures, one would also like to be able to assess the correspondence between the simulations for high flows. A full development and discussion of the results, with reference to the different aspects considered here (i.e. snowmelt vs. rainfall-dominated catchments and high flow vs. low flow regimes) would significantly enhance the contribution of this work to the scientific literature. Overall, I find this to be a valuable piece of work will be worthy of publication, once the issues raised above have been addressed. Otherwise, I have only a few additional minor comments, questions and proposed corrections as given below.

**Reply:** *The validation results were also shown for high-flow, i.e., Figure 6 in the manuscript was completed by the high-flow regime estimates.*

**Modification: Figure 6**

**Specific comments**

P1 – Abstract: Well-presented, but at this point in the manuscript I also needed a clarification as to what is meant by 'flow regime'. Perhaps you should more clearly emphasis that in your end results you are interpreting/analysing 'flow regimes' using annual hydrographs comprised of monthly averaged flows. Particularly, for the case of high flow regimes, the time unit analysed is of interest.

**Reply:** *A specification was added to the text.*

**Modification: p:1, l:3**

P2 L24-25 I don't agree with the statement that the focus on an individual characteristic "neglects the pre-conditions of low- and high-flow events", at least in terms of how most climate impact studies consider changes in this variables. For example, if one is evaluating changes in maximum daily flows based on daily simulations of 30-year periods for a present and a future climate, potentially higher flows in the period preceding the largest events are

also represented in those simulations, i.e. they are not neglected. It is true that by considering an index which targets a broader time window or discharge range you are able to more directly interpret why these changes occur, e.g. because discharge is elevated for a longer time period and is thus more susceptible to extreme precipitation. In the 'standard' one characteristic approach, this would require a further analysis. A similar argument can be made for low flow indices, i.e the antecedent conditions are simulated and can be analysed if one requires additional information as to factors responsible for the estimated changes. However, to simply state that pre-conditions are neglected is misleading and incorrect.

**Reply:** *We did by no means intend to say that the use of climate simulations does not allow for the consideration of extreme events. What we intended to say was that frequency analyses often focus on one hydrological characteristics instead of considering the flow regime as a whole. Flood frequency or low flow frequency analyses typically do not explicitly consider antecedent conditions. The sentence was rephrased to avoid confusion.*

**Modification: p:2, l:22**

P3 L15-20 As mentioned in the general comments, this paragraph needs to justify why direct stochastic simulation of discharge is used in this work. I would therefore suggest that the paragraph opens with a sentence describing what stochastic simulation is in general, and that this is followed by a more thorough discussion of the advantages and disadvantages of indirect vs. direct approaches, before you jump into a detailed discussion of options for direct simulation. It is important that you place the direct stochastic simulation method you have chosen within its broader context, and argue for why it is preferable and suitable for use here. In particular, there is a wide and growing literature on 'indirect' stochastic simulation which should be covered here with at least 2-3 sentences.

**Reply:** *We introduced the aim of stochastic simulation and added a short overview on the two different main approaches: direct and indirect. Then, we explained that we chose a direct simulation approach because it's easy to apply and works without having to progress precipitation through a hydrological model.*

**Modification: p:3, l:15-19**

P6 L17 'regional downscaling approach based on quantile mapping'. . ..more detail is needed. In particular, it is not clear in the description given in L20-L21 at what point the bias correction with quantile mapping is applied. I assume that the same data which is used for the simulations based on observed data is used for the bias correction. . ..is this correct? In addition, what is the time period of the observed data used for bias correction?

**Reply:** *The quantile mapping was applied to simulated raw model data, which were bilinearly interpolated from the 12 or 50 km RCM resolution to the observational 2 km grid. It was performed on a grid-cell-by-grid-cell basis. The calibration was performed with the observed gridded data on the period 1981-2010, which was used to drive the model in the control run. More detail on the quantile mapping procedure were provided in the updated version of the manuscript.*

**Modification: p:6, l:31-32**

P6 L18 . . ..'39 model chains (Table A1'. The choice of models used should be discussed, rather than leaving it to the reader to decipher a table in the appendix in order to determine how many different GCMs vs. RCMs vs. RCPs vs. grid resolutions are represented by the 39 simulations. The different models, etc. represented in the ensemble will indeed have an effect on both the mean values estimated and the spread about the mean, and the 39 model chains used here are only a subset of available EUROCORDEX simulations. In particular, it is unclear to me why both EUR-11 and EUR-44 grid resolutions have been used, if the climate model data are used for hydrological simulations at a 200-m grid resolution. I suspect that the EUR-44 simulations have been included to give a larger ensemble; however, this also means that the GCMRCM combinations that are available for both EUR-11 and EUR-44 are more heavily weighted in the ensemble. A brief (2-3 lines) discussion of the composition of the ensemble is therefore needed in the manuscript.

**Reply:** *The ensemble was more thoroughly described and a discussion on its composition was provided.*

**Modification: p:6, l:34-p:7, l:3**

P7 L13-14 Need to also generate stochastic series for the current 'conditions' using the hydrological model simulations based on the 39 model chains for the period 1981-2010.

**Reply:** *We derived stochastic simulations from the reference simulations derived from each of the 39 model chains. These were later on used to estimate the extreme regimes for the reference period and to compute a multi-model mean serving as a reference.*

**Modification: p:7, l:29-31**

P8 L2 – 'Long-term mean'. . .can be more specific, i.e. the mean regime over the period 1981-2010? I assume from your figures that this is aggregated by month?

**Reply:** *The long-term mean for the current conditions was computed over the period 1981-2010. The one for future conditions over the period 1971-2100. We specified that the regime varied for current and future conditions and that it was computed at a daily resolution but aggregated to a monthly scale to allow for a comparison between the univariate and FDC estimates.*

**Modification: p:8, l:28-29**

P8 L14-15 It is this 'reference' simulation which should actually be the 'control' simulation and be further used to evaluate future changes

**Reply:** *We agree and now use a multi-model mean reference simulation to derive the differences in regime estimates between future and current conditions.*

**Modification: p:9, l:11-12**

P8 L17 'were treated separately'. . ..not sure what is meant by this. Perhaps simply that the results were grouped by RCP?

**Reply:** *This was rephrased as suggested.*

**Modification: p:9, l:12-13**

P10 L5-7 Why is the seasonality so variable between the FDC, Stochastic and Univariate estimates in the rainfall-dominated Thur and Jura catchments? Doesn't this undermine the credibility of these methods for considering high flows in these and other rainfall-dominated catchments?

**Reply:** *We specified that the univariate approach produces unrealistic results in terms of seasonality, since the predictions of the monthly 100-year flows neglects the dependence between the different months. The FDC and stochastic approaches produce rather similar results excepts for the Thur catchment. While the FDC approach works with a mean seasonality, the stochastic approach uses the seasonality of one specific realization (annual hydrograph), whose discharge sum corresponds to the 100-year discharge sum.*

**Modification: p:10, l:2-3, p:18, l:5-10**

P10 L10-12 Given my comment above, I am also curious as to what Fig. A1 looks like for high-flow regime estimates? In particular, the stochastic results for Jura in Fig. 5 suggest that the method doesn't perform better than the univariate approach in that case.

**Reply:** *The stochastic approach is not supposed to exactly reproduce the mean seasonality reflected by the FDC approach. While the seasonality generated by the univariate approach is something completely artificial, the seasonality produced by the stochastic approach could have been observed. Figure A1 was also produced for the high-flow estimates and is provided here for completeness.*

[Figure]

**Figure 2: Comparison of the 100-year high-flow estimates, univariate, FDC, and stochastic with stochastically generated hydrographs (orange lines). The legend is equivalent to the one provided for Figure A1 in the manuscript.**

P11 Sec 3 Would also like to see a comparison of the climate simulations and the control simulation for high flows, similar to that shown for Fig 6 for low flow regimes.

**Reply:** *Figure 6 was extended by the high-flow regimes.*

**Modification: Figure 6**

P12 L10-11: Changes in mean flow of up to 50%, etc. These estimates are not reliable because you are comparing the results of simulations based on climate model data (future) with those based on observations (current), and thus are also including some the error illustrated in Figure 6 in your estimates.

**Reply:** *We newly computed the differences between current and future regime estimates by using the multi-model mean derived from the 39 reference simulations as a reference estimate. The text was adjusted accordingly.*

**Modification: 3.2 current and future low-flow estimates**

P17 Section 4.1: In addition to discussing the overall merits of the two methods, it would also be useful to see a discussion of their relative performance for high vs. low flow regimes and for rainfall-dominated vs. snowmelt-dominated catchments. I also find that the second paragraph in this section is too general and should focus on discussing and expanding the results you have presented in more depth. It would also be useful, for example, if you could highlight aspects of the methods you have used which are better for quantifying changes in

flow regimes useful for management purposes, relative to those commonly used for climate impact studies.

**Reply:** *The differences between the estimates derived using the FDC approach and those derived using the stochastic approach were discussed in more detail. However, it is difficult to say which one is the "better" method since the "true" 100-regime estimate is not known.*

**Modification: p:18, l:5-10**

P18 L15-16: 'Both were found to provide realistic, mutually-agreeing results'. I think that this is bit overstated. In particular, the results for the high flow regimes for the 19 regions (Fig. 10) show a similar direction of change (in most cases), but the magnitude is in some cases much higher with the Stochastic method, and this would have significant implications for their application in practice.

**Reply:** *The statement was weakened by replacing mutually-agreeing with similar.*

**Modification: p:15, l:4**

**Technical comments**

P1 – Keypoints: (L22) – 'are changing' should be 'will change', i.e. you have not examined patterns of change under current conditions, which is what the English used here implies

**Reply:** *The keypoint was rephrased.*

**Modification: Keypoint**

P2 L17-18: Considering rephrasing, for example to: For planning purposes and river basin management, however, estimates not only for normal conditions but also for extreme conditions are needed.

**Reply:** *The sentence was rephrased.*

**Modification: p:2, l:17-18**

P4 L3. . .should add 'under the current climate' after 'regime.'

**Reply:** *We specified this.*

**Modification: p:4, l:9**

P12 L4 Replace 'expressed' with 'pronounced'

**Reply:** *Expressed will be replaced by pronounced.*

**Modification: p:13, l:6**

P14 Fig 8; P16 Fig10 caption: In line 3 should be 'The top three rows show relative changes, and the bottom two rows show changes in months'

**Reply:** *The captions were adjusted.*

**Modification: Captions 8 and 10**

P18 L8-9: Replace 'We here showed' with 'We have shown here'

**Reply:** *The sentence was rephrased.*

**Modification: p:19, l:15-16**

**References used in the answers to the reviewers**

Hosking, J.R.M., 1994. The four-parameter kappa distribution. IBM J. Res. Dev. 38, 251–258.

[revised manuscript text omitted]

---

## Author Response (AR2)

**Dear Prof. Bronstert,**

We thank you for reconsidering the manuscript for publication in HESS. Below, we address the technical point risen by reviewer two and state how we addressed it. Our replies to the reviewers' comments are written in blue and italic to distinct them from the reviewers' comments.

On the behalf of all co-authors,

Yours sincerely,

Manuela Brunner

**Reviewer 2**

Based on the revised Figures 7 – 10, it appears that the authors have addressed the major concern raised in my previous review, i.e. the need for a comparison of future simulations with simulations for the current period, both of which should be based on climate model data. In the previous version of this paper, the 'current' period was modelled using observed meteorological data and used for comparisons with the future period. As I understand it from the authors' reply to my review (and please correct me if I have misunderstood), this has now been replaced in the figures by the multi-model mean value from the 39 climate simulations for the reference period for comparison with the full distribution of results for the future period. In other words, a mean value estimated from the distribution of 'reference' simulations shown in Figure 6 has been used in Figures 7-10. The effects of this change is most noticeable in the significant reduction in the variability in the change in the mean discharge between catchments, shown in Figure 8 and 10. That you have chosen to use the multi-model mean for these comparisons rather than the full distribution for the reference period needs to be mentioned in the text; otherwise, it is unclear as to what 'Current' refers to in the legends for fig. 7 and 9, and what the baseline (i.e. 0 value) is for Figures 8 and 10. In addition, Fig. 10 needs be adjusted so that the full range of values is shown for the box plots in the first row. Other than these two very minor changes, I have no further comments.

*Reply: Thank you very much for reviewing the manuscript again. It is correct that 'current' now refers to the multi-model mean of the reference simulations. We specify on p.9 (l7-10) that the 39 GCM-RCM combinations were used to derive a multi-model mean, which served as a reference for determining changes between current and future conditions. We added this to the captions of Figures 7-10.*
*We tried to choose a common plotting axis for displaying the relative changes in low-flow regimes (Figure 8) and high-flow regimes (Figure 10). Adjusting the y-axis of the first row of Figure 10 would put the different subfigures on different scales, which we try to avoid here. Adjusting all the y-axes jointly would be an option but such an increase in scale has the disadvantage of reducing the size of the small boxplots even more, which would hide some details. We therefore prefer to keep the scales of Figures 8 and 10 as they are.*
**Modification: p.14, l:1 and captions of Figures 7-10.**